# Towards Enhanced Controllability of Diffusion Models

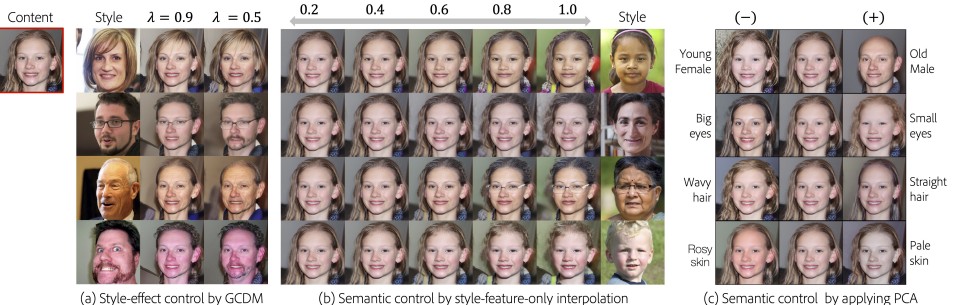

Figure 1: Diverse results from our proposed methods given single content image (the leftmost). Controlling $\lambda$ in our proposed sampling method (GCDM) results in different magnitudes of style translation in (a). While fixing the content feature, applying convex combinations between the style features of the content and the style images shows smooth interpolations as shown in (b). PCA on the learned style code gives disentangled attribute-specific manipulation directions in (c). Detailed experiment setups and more results are in Sections F.1, F.2 and in Fig. 18 in the supplementary.

## ABSTRACT

As Diffusion Models have shown remarkable capabilities in generating images, the controllability of Diffusion Models has received much attention. However, there is still room for improvement of controllability in some aspects, such as feature disentanglement of Diffusion Models for extended editability and composing multiple conditions naturally. In this paper, we present three methods that can be used in either training or sampling to enhance the controllability of Diffusion Models. Concisely, we train Diffusion Models conditioned on two latent codes, a spatial content mask, and a flattened style embedding. We rely on the inductive bias of the progressive denoising process of Diffusion Models to encode pose/layout information in the spatial structure mask and semantic/style information in the style code. We also propose two generic sampling techniques for improving controllability. First, we extend Composable Diffusion Models to allow for some dependence between conditional inputs, to improve the quality of generations while also providing control over the amount of guidance from each condition and their joint distribution. Second, we propose timestep-dependent weight scheduling for content and style latents to further improve the translations. We observe better controllability compared to existing methods and show that with our proposed methods, Diffusion Models can be used for effective image manipulation and image translation.

## 1 INTRODUCTION

Improving controllability of generative models has been one of the most prominent research topics in past few years, e.g., GANs (Goodfellow et al., 2014; Härkönen et al., 2020), VAE (Kingma & Welling, 2013; Bouchacourt et al., 2018), Flow-based Models (Dinh et al., 2017; Esser et al., 2020), Masked Generative Transformers (Chang et al., 2023), and Autoregressive Models (Yu et al., 2023). The enhanced controls are useful for many practical applications such as Image Synthesis (Park

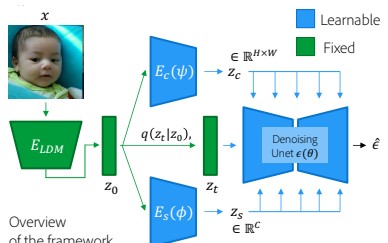

Overview
of the framework

Figure 2: overview of our proposed framework. We first obtain $z_0$ from the pretrained Autoencoder (Esser et al., 2021), which is the actual input for the LDM (Rombach et al., 2022). The external encoders $E_c(\psi)$ and $E_s(\phi)$ and the denoising UNet $\epsilon(\theta)$ are trained together without any additional objectives.

et al., 2020), Domain Adaptation (Hoffman et al., 2018), Style Transfer (Huang et al., 2018; Lee et al., 2018) and Interpretability (Lang et al., 2021) to name a few.

Recently, Diffusion Models (Sohl-Dickstein et al., 2015; Ho et al., 2020) have gained much attention due to their impressive performance in image generation (Dhariwal & Nichol, 2021; Ramesh et al., 2022; Rombach et al., 2022) and likelihood estimation (Nichol & Dhariwal, 2021). Even though there have been many research papers on extending controllability of Diffusion Models, it has relatively remained underexplored how to disentangle the latent space of diffusion models, and how to combine the multiple conditions naturally during the sampling in a controllable way.

Indeed, the topic of generative models with multiple external disentangled latent spaces has been widely explored in GANs (Park et al., 2020; Huang et al., 2018; Lee et al., 2018; Kwon & Ye, 2021). A common theme across such methods is to learn a structure/content code to capture the underlying structure (e.g., facial shape and pose in face images) and a texture/style code to capture global semantic information (e.g. visual appearance, color, hair style etc.). Similar approaches have been tried in Diffusion Models in (Kwon & Ye, 2022; Preechakul et al., 2022), however, these techniques do not learn multiple controllable latent spaces.

In this paper, we propose a novel framework as shown in Fig. 2 to effectively learn two latent spaces to enhance controllability in diffusion models. Inspired by (Park et al., 2020; Kwon & Ye, 2021) we add a *Content Encoder* that learns a spatial layout mask and a *Style Encoder* that outputs a flattened semantic code to condition the diffusion model during training (Section 3.1). The content and style codes are injected differently into the UNet (Ronneberger et al., 2015) to ensure they encode different semantic factors of an image.

Though decomposing content and style information from an image enables better controllability, enforcing independence between the codes during sampling may not always be ideal. For example, *face structure* (e.g. square or round face) that is ideally encoded in the content code and *gender* (e.g. male or female) an attribute encoded in the style code (Park et al., 2020), may not be independent and treating them as such might lead to unnatural compositions (Fig. 3). However, an existing method Composable Diffusion Models (CDM) (Liu et al., 2022) assumes conditioning inputs are independent and hence shows unnatural compositions for certain prompts like 'a flower' and 'a bird' (see Fig.6 in (Liu et al., 2022)). We extend the formulation in (Liu et al., 2022) and propose *Generalized Composable Diffusion Models* (GCDM) to support compositions during inference when the conditional inputs are not necessarily independent (Section 3.3). This also provides the ability to control the amount of information from content, style, and their joint conditioning separately during sampling. We observe significantly better translations with GCDM and also show improved compositions in Stable Diffusion compared to CDM (Fig. 5).

In addition, we leverage the inductive bias (Balaji et al., 2022; Choi et al., 2021; 2022) of Diffusion Models that learns low-frequency layout information in earlier steps and high-frequency or imperceptible details in the later steps of the reverse diffusion process, to further improve results. We use a predefined controllable timestep-dependent weight schedule to compose the content and style codes during generation. This simulates the mixture of denoising experts proposed in (Balaji et al., 2022) by virtue of varying the conditional information at different timesteps during inference.

# 2 PRELIMINARIES AND RELATED WORKS

## 2.1 DIFFUSION MODELS

Diffusion Models (Sohl-Dickstein et al., 2015) like DDPM (Ho et al., 2020) showed impressive image generation and likelihood estimation but had a computationally expensive sampling procedure. DDIM (Song et al., 2020) reduced the sampling time by deriving a non-Markovian variant of DDPM. Similarly, ImprovedDDPM (Nichol & Dhariwal, 2021) also improved sampling speed and proposed to learn the variance schedule that was fixed in previous works to enhance mode coverage.

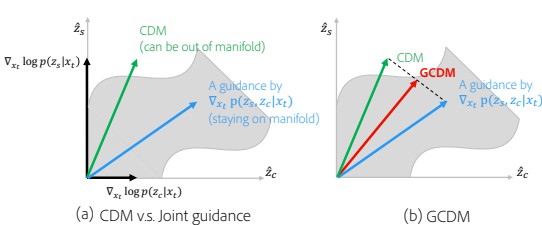

Figure 3: Conceptual illustration of CDM and GCDM. (a) The result based on CDM can be outside of manifold while the joint guidance stays on manifold. (b) GCDM trades off between the independent guidance provided by CDM (stronger effects of the condition) and the joint guidance (more realistic). Please see Fig. 5, 6 (main paper) and Fig. 16 (supplementary).

LDM (Rombach et al., 2022) used a pretrained autoencoder (Esser et al., 2021) to train a diffusion model on the learned latent space, reducing time and memory complexity without loss in quality. More descriptions are provided in Section D in the supplementary.

## 2.2 CONTROLLABILITY IN DIFFUSION MODELS

**Guidance:**
Some recent works have explored modeling the conditional density $p(x_t|c)$ for controllability. Dhariwal et al. (Dhariwal & Nichol, 2021) proposed to use a pretrained classifier but finetuning a classifier that estimates gradients from noisy images, which increases the complexity of the overall process (Ho & Salimans, 2022). Ho et al. (Ho & Salimans, 2022) proposed to use an implicit classifier while Composable Diffusion Models (Liu et al., 2022) (CDM) extend the classifier-free guidance approach to work with multiple conditions assuming conditional independence. Though guidance approaches help control the generation, they do not offer fine-grained controllability or support applications such as reference-based image translation.

**Conditional Diffusion Models:**
Conditional Diffusion Models have been explored in diverse applications showing state-of-the-art performance in text-to-image generation (DALLE2 (Ramesh et al., 2022), Imagen (Saharia et al., 2022), Parti (Yu et al., 2022)). These methods use pretrained embeddings (e.g., CLIP) that support interpolation but not further editability. Instructpix2pix (Brooks et al., 2023) proposed to generate synthetic paired data via pretrained GPT-3 (Brown et al., 2020) and StableDiffusion, with which conditional Diffusion Models are trained. DiffAE (Preechakul et al., 2022) proposed to learn a semantic space that has nice properties making it suitable for image manipulation. However, a single latent space capturing all the information makes it difficult to isolate attributes to manipulate. Recently, ControlNet and T2Iadapter (Mou et al., 2023; Zhang & Agrawala, 2023) showed impressive performance in conditioning image generation. They use additional auxiliary networks and layers that are trained to encode structure into pretrained text2image Diffusion Models. However, our architecture is particular to reference-based image translation, the proposed GCDM and timestep scheduling are generic and applicable to any multi-conditioned Diffusion Models beyond image translation.

**Inference only Editing:**
SDEdit (Meng et al., 2021) enables structure-preserving edits while Prompt-to-prompt (Hertz et al., 2022) modifies the attention maps from cross-attention layers to add, remove, or reweigh the importance of an object in an image. DiffusionCLIP (Kim et al., 2022), Imagic (Kawar et al., 2022) and Unitune (Valevski et al., 2022) propose optimization-based techniques for text-based image editing. Textual Inversion (Gal et al., 2022) and DreamBooth (Ruiz et al., 2022) finetune pretrained models using a few reference images to get personalized models. Though the above techniques are helpful with editing, most of these methods require computationally expensive optimization, modify the weights of pretrained model for each sample, and/or don't support fine-grained controllability for reference-based image translation. The closest related work to ours is DiffuseIT (Kwon & Ye, 2022). They enabled reference and text-guided image translation by leveraging Dino-VIT (Caron et al., 2021) to encode content and style. However, their approach requires costly optimization during inference and doesn't support controlling the final generation.

## 3 PROPOSED METHOD

Our framework is based on the LDM (Rombach et al., 2022) architecture as it is faster to train and sample from, compared to pixel-based diffusion models. Let $x$ be an input image and $E_{LDM}$ and $D_{LDM}$ be the pretrained and fixed encoder and decoder respectively. The actual input space for our diffusion model is the low-dimensional latent space $z = E_{LDM}(x)$. The output of the

reverse diffusion process is the low dimensional latent $\hat{z}_0$ which is then passed through the pretrained decoder as $x = D_{LDM}(\hat{z}_0)$ to get the final image $\hat{x}_0$.

## 3.1 Learning Content and Style Latent spaces

Inspired by DiffAE (Preechakul et al., 2022) and similar approaches in GANs (Kwon & Ye, 2021), we introduce a content encoder $E_c(\,\cdot\,;\psi)$ and a style encoder $E_s(\,\cdot\,;\phi)$ in our framework as shown in Fig. 2. The objective for training is formulated as:

$$\min_{\theta,\psi,\phi} \mathbb{E}_{z_0,\epsilon_t} \left[ \|\epsilon_t - \epsilon(z_t, t, E_c(z_0;\psi), E_s(z_0;\phi);\theta)\|_2^2 \right],$$

where $z_t$ is from the forward process, i.e., $z_t = q(z_t|z_0)$. To ensure that the encoders capture different semantic factors of an image, we design the shape of $z_s$ and $z_c$ asymmetrically as done in (Park et al., 2020; Tumanyan et al., 2022; Huang et al., 2018; Lee et al., 2018; Kwon & Ye, 2021; Cho et al., 2019). The content encoder $E_c(z_0;\psi)$ outputs a spatial layout mask $z_c \in \mathbb{R}^{1 \times \frac{h}{4} \times \frac{w}{4}}$ where $w$ and $h$ are the width and height of $z_0$ latent. In contrast, $E_s(z_0;\phi)$ outputs $z_s \in \mathbb{R}^{512 \times 1 \times 1}$ after global average pool layer to capture global high-level semantics. At each layer of the denoising UNet $\epsilon(\,\cdot\,;\theta)$, the style code $z_s$ is applied using channel-wise affine transformation while the content code $z_c$ is applied in a spatial manner. In specific, let $\odot$ denote the element-wise product (i.e., Hadamard product), let $\otimes$ denote outer product, and let $\mathbf{1}$ denote all one's vector/matrix where subscript denotes dimensionality. We define a variant of adaptive group normalization layer (AdaGN) in the UNet as:

$$\begin{aligned} \text{AdaGN}(h^\ell) = & [\mathbf{1}_C \otimes (\mathbf{1}_{H,W} + t_1 \varphi^\ell(z_c))] \odot [(\mathbf{1}_C + \zeta^\ell(z_s)) \otimes \mathbf{1}_{H,W}] \\ & \odot [[(\mathbf{1}_C + \boldsymbol{t}_2) \otimes \mathbf{1}_{H,W}] \odot \text{GN}(h^\ell) + [\boldsymbol{t}_3 \otimes \mathbf{1}_{H,W}]], \end{aligned} \quad (1)$$

where $\varphi^\ell$ is the spatial-wise content-specific term operating down or upsampling at $\ell$-th layer to make the dimensions of $\varphi^\ell(z_c)$ and $h^\ell$ match. $\zeta^\ell$ is the channel-wise style-specific term which is implemented as an MLP layer. $h^\ell$ is (hidden) input to the $\ell$-th layer, and $t_1$, $t_2$ and $t_3$ are timestep-specific adjustment terms inspired by Eq. 7 in DiffAE (Preechakul et al., 2022). The $(1 + \_)$ structure reveals the residual architecture of the content, style, and timestep modifications that maintain the input if all adjustment terms are zero. When $C, H, W$ are all 1 (i.e., the scalar case), the equation can be simplified to reveal the basic structure as : $(1 + t_1 \varphi(z_c))(1 + \zeta(z_s))((1 + t_2)h + t_3)$.

## 3.2 Timestep Scheduling for Conditioning

It has been observed in (Choi et al., 2021; 2022; Balaji et al., 2022) that low-frequency information, i.e., coarse features such as pose and facial shape are learned in the earlier timesteps (e.g., $0 < \text{SNR(t)} < 10^{-2}$) while high-frequency information such as fine-grained features and imperceptible details are encoded in later timesteps (e.g., $10^0 < \text{SNR(t)} < 10^4$) in the reverse diffusion process.

Inspired by this, we introduce a weight scheduler for $z_c$ and $z_s$ that determines how much the content and the style conditions are applied to the denoising networks. We use the following schedule:

$$w_c(t) = \frac{1}{1 + \exp\left(-a(t - b)\right)}, \quad w_s(t) = \frac{1}{1 + \exp\left(-a(-t + b)\right)}, \quad (2)$$

where $a$ is a coefficient for determining how many timesteps content and style are jointly provided while $b$ indicates the timestep at which $w_s(t) \geq w_c(t)$. We also tried a simple linear weighting and a constant schedule but observed that the proposed schedule gave consistently better results (examples are provided in Section G.2 in the supplementary). When applying, we use weighted form, denoted $\bar{\varphi}$ and $\bar{\zeta}$, of the style and content functions during sampling. They are defined as $\bar{\varphi}(z_c, t) := w_c(t)\varphi(z_c)$ and $\bar{\zeta}(z_s, t) := w_s(t)\zeta(z_s)$, which respectively replace $\varphi$ and $\zeta$ in Eq. 1.

We additionally evaluate using timestep scheduling during training. It is an interesting future direction showing better decomposition of content and style (Section G.1 in the supplementary).

## 3.3 Generalized Composable Diffusion Models (GCDM)

As mentioned in Section 1, CDM has an inherent limitation that conditional independence is assumed (i.e., $C_1 \perp C_2 | X_t$), which may not always hold in practice. Incorporating the joint component into CDM formulation possibly yields a better composition of seemingly irrelevant objects in

the real world, e.g., $c_1 =$ 'an octopus' and $c_2 =$ 'a pyramid', by finding the real manifold that both objects can be naturally placed together. Thus, we propose GCDM which can potentially improve the composition of multiple conditions.

Furthermore, GCDM formulation can enhance controllability over the generation enabling extended control over CDM. The conceptual benefit of GCDM over CDM can be understood by Fig. 3. (a) shows an example that the content and the style guidances from CDM generate unrealistic samples because the combined guidance is outside the manifold. On the contrary, the joint guidance helps keep the generation within the manifold. (b) visualizes the proposed GCDM which can be seen as a linear interpolation between CDM and the joint guidance. GCDM has the added advantage of enabling separate controls for $c_1$ (e.g., style), $c_2$ (e.g., content), and *realism*.

**Definition 3.1** (Generalized Composable Diffusion Models (GCDM)). The score function of GCDM is the unconditional score function plus a convex combination of joint and independent guidance terms formalized as:

$$\nabla_{x_t} \log \tilde{p}_{\alpha,\lambda,\beta_1,\beta_2}(x_t|c_1,c_2) \triangleq \epsilon(x_t,t) + \alpha\Big[\lambda(\underbrace{\epsilon(x_t,t,c_1,c_2) - \epsilon(x_t,t)}_{\nabla_{x_t}\log p(c_1,c_2|x_t)}) \tag{3}$$
$$+ (1-\lambda)\sum_{i=\{1,2\}}\beta_i\Big(\underbrace{\epsilon(x_t,t,c_i) - \epsilon(x_t,t)}_{\nabla_{x_t}\log p(c_i|x_t)}\Big)\Big],$$

where $\alpha \geq 0$ controls the strength of conditioning, $\lambda \in [0,1]$ controls the trade-off between joint and independent conditioning, and $\beta_i$ controls the weight for the $i$-th condition under the constraint that $\sum_i \beta_i = 1$.

Note that the implicit classifiers, e.g., $\nabla_{x_t} \log p(c_i|x_t)$, play a role in guiding $x_t$ to be close to the corresponding condition. The GCDM formulation $\tilde{p}$ in Definition 3.1 stems from the idea of extending controllability of naive joint conditioning $\nabla_{x_t} \log p(x_t|c_1,c_2)$ to mixed conditioning between joint and independent guidance. Similar to previous studies (Ho & Salimans, 2021; Liu et al., 2022), the guidance terms (i.e., the implicit classifiers) to be controlled are derived from reformulating the joint conditioning by simple Bayes rule. Please see Section B in the Supplementary for the derivations.

We next show some of the interesting features of GCDM. First, GCDM generalizes simple joint guidance, CDM, and Classifier-Free Guidance (Ho & Salimans, 2021) (CFG).

**Proposition 3.2** (GCDM Generalizes Joint Guidance, CDM and CFG). *If $\lambda = 1$, then GCDM simplifies to joint guidance:*

$$\nabla_{x_t} \log \tilde{p}_{\lambda=1}(x_t|c_1,c_2) = \underbrace{\epsilon(x_t,t) + \alpha(\epsilon(x_t,t,c_1,c_2) - \epsilon(x_t,t))}_{Joint\ Guidance} = \nabla_{x_t}\log p(x_t|c_1,c_2). \tag{4}$$

*If $\lambda = 0$, then GCDM simplifies to CDM:*

$$\nabla_{x_t} \log \tilde{p}_{\lambda=0}(x_t|c_1,c_2) = \epsilon(x_t,t) + \alpha\underbrace{\Big[\sum_{i=\{1,2\}}\beta_i(\epsilon(x_t,t,c_i) - \epsilon(x_t,t))\Big]}_{CDM}. \tag{5}$$

*If $\lambda = 0$ and $\beta_2 = 0$, then GCDM simplifies to CFG:*

$$\nabla_{x_t} \log \tilde{p}_{\lambda=0,\beta_2=0}(x_t|c_1,c_2) = \underbrace{\epsilon(x_t,t) + \alpha\beta_1(\epsilon(x_t,t,c_1) - \epsilon(x_t,t))}_{CFG}. \tag{6}$$

The proof is simple from inspection of the GCDM definition. Second, the GCDM PDF $\tilde{p}$ from Definition 3.1 is proportional to a nested geometric average of different conditional distributions.

**Corollary 3.3.** *The GCDM distribution $\tilde{p}$ is proportional to nested geometric averages of conditional distributions of $x_t$:*

$$\tilde{p}_{\alpha,\lambda,\beta_1,\beta_2}(x_t|c_1,c_2) \propto p(x_t)^{(1-\alpha)}\Big[p(x_t|c_1,c_2)^{\lambda}\Big(p(x_t|c_1)^{\beta_1}p(x_t|c_2)^{(1-\beta_1)}\Big)^{(1-\lambda)}\Big]^{\alpha}. \tag{7}$$

The outermost geometric average is between an unconditional and conditional model. Then inside we have the geometric average of the joint and independent conditional, and finally inside the independent conditional we have a geometric average of the independent conditionals. The derivation is provided in Section C in the supplementary.

Note that GCDM and timestep scheduling are generic sampling techniques for Diffusion Models that can also be applied to other tasks beyond image translation (Fig. 5). A thorough investigation of the effect of the various hyperparameters is provided in Section E.1 in the supplementary.

## 4 EXPERIMENTS

We comprehensively evaluate the proposed model on image-to-image translation and additionally show qualitative examples of GCDM and CDM on text-to-image composition with stable diffusion. Implementation details are provided in Section E.

### 4.1 EXPERIMENTAL SETUP

**Datasets**
We train different models on the commonly used datasets such as AFHQ (Choi et al., 2020), FFHQ (Karras et al., 2019) and LSUN-church (Yu et al., 2015).

**Baselines**
**DiffuseIT:** The most similar work to ours based on diffusion models is DiffuseIT (Kwon & Ye, 2022) that tackles the same problem formulation. We compare our results with DiffuseIT using their pretrained model and default parameters.
**DiffAE+SDEdit:** Since Diffusion Autoencoder (Preechakul et al., 2022) does not directly support image-to-image translation, we combine that with SDEdit (Meng et al., 2021). The input image for the reverse process is $x_{600}$ (chosen empirically) obtained as $q(x_{600}|x_c)$ by running the forward process on the content image. The semantic feature $z_{sem}$ from the semantic encoder of DiffAE is used given the style image $x_s$.
**DiffAE+MagicMix:** We also combine MagixMix (Liew et al., 2022) with DiffAE. Similar to DiffAE+SDEdit, this model takes $x_{600}$ from $x_c$ as input and $z_{sem}$ from $x_s$ as conditioning. Additionally, at each timestep, the approximated previous timestep $\hat{x}_{t-1}$ is combined with $x_{t-1}$ from the content image $x_c$, i.e., $\hat{x}_{t-1} = v\hat{x}_{t-1} + (1-v)q(x_{t-1}|x_c)$. For this experiment, $v = 0.5$ is used and the noise mixing technique is applied between $t = [600, 300]$.
**SAE:** Swapping Autoencoder (Park et al., 2020) based on GAN (Goodfellow et al., 2014) is also evaluated. Since the available pretrained model is on a resolution of 512, we resize the generated results to 256 for a fair comparison.

**Evaluation Metrics**
**FID:** We use the commonly used Fréchet inception distance (FID) (Heusel et al., 2017) to ensure the generated samples are realistic. We follow the protocol proposed in (Choi et al., 2020) for reference-based image translation. To obtain statistics from generated images, 2000 test samples are used as the content images, and five randomly chosen images from the rest of the test set are used as style images for each content image to generate 10,000 synthetic images.
**LPIPS:** Even though FID evaluates the realism of the generations, the model could use just content and ignore style (or vice versa) and still get good FID. Following (Choi et al., 2020), we use LPIPS score obtained by measuring the feature distances between pairs of synthetic images generated from the same content image but with different style images. **Higher LPIPS indicates more diverse results**. It is ideal for the model to tradeoff between LPIPS and FID, i.e., incorporate enough style information from different style images for the same content image (increasing LPIPS) but without going out of the real distribution (decreasing FID).

### 4.2 COMPARISON WITH EXISTING WORKS

**Qualitative Results.**
Fig. 4 visually shows example generations from different techniques. We observe that DiffAE+SDEdit loses content information while DiffAE+MagicMix generates unnatural images that naively combine the two images. This indicates that a single latent space even with additional

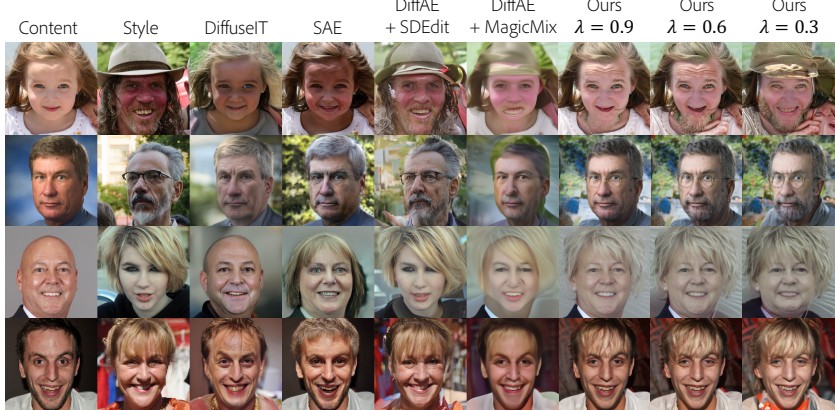

Figure 4: Comparison of the proposed model with baselines on FFHQ dataset. Our method generates more realistic combinations of the content and style images with better controllability.

techniques such as SDEdit and MagicMix is not suitable for reference-based image translation. DiffuseIT and SAE models maintain more content information but do not transfer enough information from the style image and have no control over the amount of information transferred from style.

An important benefit of our proposed method is better controllability. First of all, by manipulating $\lambda$, we can control how much joint guidance is applied. In Fig. 4, decreasing $\lambda$ indirectly increases the effect of style from the style image when $\beta_c = 0$ and $\beta_s = 1$, where $\beta_c$ and $\beta_s$ are the weights for each conditional guidance. It is because the smaller $\lambda$ brings more information from the style guidance (ref. $1 - \lambda$ term in Eq. 3). For example, the man on the second row has more wrinkles and a beard as $\lambda$ decreases. Second, given a fixed value of $\lambda$, we can control the amount of the content and the style guidance by controlling $\beta_c$ and $\beta_s$ as shown in Fig. 6. More examples showing the superior performance of our method in controllability are provided in Fig. 1 in the supplementary.

Table 1: Quantitative comparison using FID and LPIPS on FFHQ dataset.

|  | DiffuseIT | SAE | DiffAE+SDEdit | DiffAE+MagicMix | Ours($\lambda = 0.9$) | Ours($\lambda = 0.6$) | Ours($\lambda = 0.3$) |
|---|---|---|---|---|---|---|---|
| FID | 29.99 | 25.06 | 26.63 | 84.55 | **11.99** | 13.40 | 15.45 |
| LPIPS | 0.47 | 0.39 | 0.64 | 0.41 | 0.34 | 0.42 | 0.49 |

**Quantitative Results.**
Table 1 shows the quantitative comparison in terms of FID and LPIPS metrics on FFHQ dataset. Our variants generate images that are realistic as indicated by the lowest FID scores compared with other models while also performing better on diversity as measured by the highest LPIPS except for DiffAE+SDEdit method. However, DiffAE+SDEdit does not show a meaningful translation of style onto the content image. DiffAE+MagicMix shows the worst performance because of its unrealistic generation. SAE and DiffuseIT show lower LPIPS scores than ours, indicating that they transfer relatively little information from the style image onto the generated samples (i.e., less diverse). We can also observe that increasing $\lambda$ (when $\beta_c = 0$ and $\beta_s = 1$) makes LPIPS worse while improving FID. In other words, the stronger the joint guidance is the more realistic but less diverse the generations. This verifies our assumption in Fig. 3 that the joint component has an effect of pushing the generations into the real manifold.

## 4.3 EFFECT OF GCDM AND TIMESTEP SCHEDULING

Table 2: FID comparisons between SAE and our model with CDM and GCDM on AFHQ dataset.

|  | SAE | CDM | GCDM ($\lambda = 0.9$) | GCDM ($\lambda = 1.0$) |
|---|---|---|---|---|
| FID | 9.29 | 10.57 | 9.75 | 8.58 |
| LPIPS | 0.45 | 0.59 | 0.59 | 0.57 |

Table 3: Comparisons between CDM and GCDM in FFHQ. Best method without timestep scheduling is highlighted in bold and with scheduling is highlighted with *.

|  | w/o schedule | | w/ schedule | | |
|---|---|---|---|---|---|
|  | CDM | GCDM | CDM | GCDM ($\beta_c = 1$) | GCDM ($\beta_s = 1$) |
| FID | 21.43 | **14.46** | 10.50 | 10.21* | 10.61 |
| LPIPS | 0.47 | **0.51** | 0.31 | 0.28 | 0.33* |

We compare SAE (Park et al., 2020) (the best performing baseline) and ours with CDM and GCDM on AFHQ dataset in Table 2. The joint guidance ($\lambda = 1$) gets the lowest FID indicating that the generations are more realistic as it pulls the guided results to be within the real data manifold. We

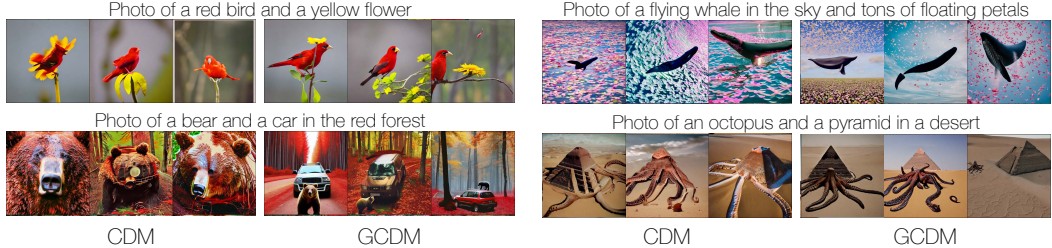

Figure 5: GCDM vs CDM for text-to-image generation with Stable Diffusion. We can observe that CDM generates unnatural images (e.g., blending two objects) that may be out of the real manifold while GCDM ensures realistic generations (e.g., combining two objects in a realistic way)

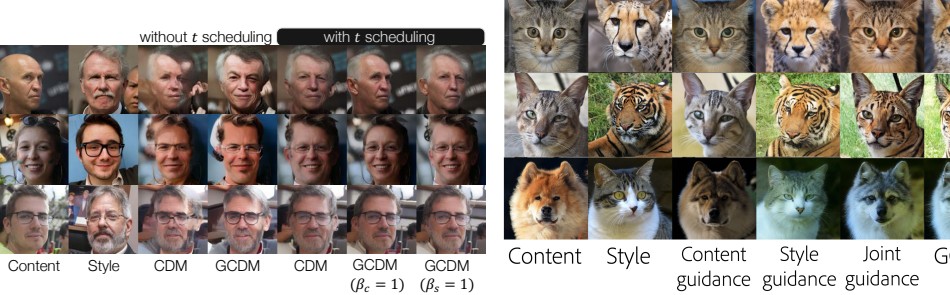

Figure 6: Timestep scheduling improved the results of both CDM and GCDM and gives the best results when combined with GCDM.

Figure 7: Visualization of the effect of each guidance term on generation. $x_T$ is randomly sampled.

can also see that GCDM can be thought of as interpolating between CDM and the joint guidance since FID for GCDM ($\lambda = 0.9$) is in between the joint and CDM. By comparing LPIPS and FID of the variants of GCDM, we can see that the outputs become less diverse as realism is increased. SAE shows worse performance than ours in terms of both diversity and realism. The qualitative comparisons can be found in Fig. 16 in the supplementary.

**Generalizability of GCDM.**
We also compare the performance of CDM and GCDM in composing text prompts for text-to-image generation using Stable Diffusion V2(Rombach et al., 2022) in Fig. 5. The phrases before and after 'and' are used as $c_1$ and $c_2$. The full sentence is used to represent joint conditioning.

As shown in Fig. 5, CDM tends to fail in composing multiple conditions if both conditions contain object information. For example, *the red bird* and *the yellow flower* are merged in most cases. On the other hand, GCDM consistently shows better compositions in the generated images. This emphasizes that GCDM is a generalized formulation for composing multiple conditioning inputs providing more control to the user in terms of realism and diversity as illustrated in Fig. 3. Additional results comparing CDM, GCDM (joint only), and GCDM can be found in Fig. 20.

**Effect of Timestep Scheduling.**
To more carefully analyze the effect of timestep scheduling when combined with GCDM or CDM, we alter the timestep scheduling so that there is at least a 0.1 weight on style or content. Specifically, we change the upper and lower bounds of the sigmoid to be 0.1 and 0.9 in Eq. 2, e.g., $w'_c(t) = 0.8w_c(t) + 0.1$. The results can be seen in Table 3 and Fig. 6. Without timestep scheduling, GCDM shows better performance in both FID (realism) and LPIPS (diversity) than CDM. Combined with timestep scheduling, both CDM and GCDM show meaningful improvements in FID in exchange for losing diversity. This is because timestep scheduling improves content identity preservation. Additionally, timestep scheduling with GCDM variants shows better FID or LPIPS than CDM depending on the strength of guidance terms showing varied control over the generations.

## 4.4 ANALYSIS AND DISCUSSION

In this section, we analyze each of the components of our framework using AFHQ and LSUN-church dataset and aim to better understand the content and style latent spaces. Further analysis and results on PCA, latent interpolation, and KNN are in Section F.1, F.2 and F.3 in the supplementary.

**Visualization of Each Guidance Term.**
The proposed GCDM in Section 3.3 has guidance from three terms, the joint conditioning and

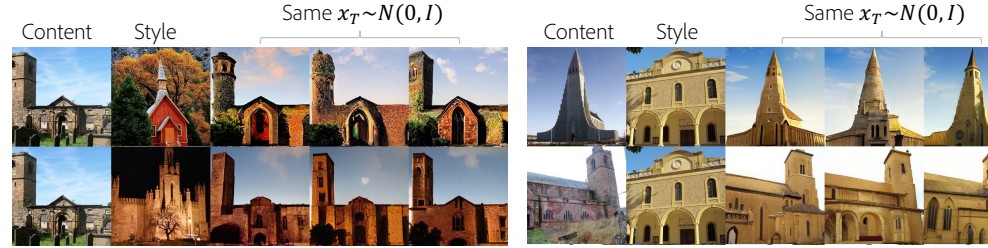

Figure 8: Reference-based image translation results on LSUN-church.

style and content conditionings separately. Fig. 7 shows a comparison of the effect of these terms. From the content guidance (column 3), it can be seen that the generated animals are not exactly the same as the content image but have the exact same structure and pose. Similarly, when only style guidance is used (column 4), the pose is random while the style such as color and fur corresponds to the style image. From columns 5-6, it can be observed that the GCDM results have more semantic information from the style than the results by simply using the joint guidance.

**Classifier-based comparisons.**
To further understand what kind of attributes are encoded in style and content latent spaces, we use pretrained classifiers to predict the attributes of translated images and compare them with the original style and content images. We sample 2000 random images

Table 4: Classifier-based comparisons in FFHQ.

| Probability Att. is Equal (%) | $x_c$ | | | $x_s$ | | |
|---|---|---|---|---|---|---|
| | Gender | Age | Race | Gender | Age | Race |
| SAE | 65.95 | 62.36 | 50.40 | 34.05 | 26.40 | 27.91 |
| Ours ($\lambda = 0.9$) | 65.14 | 53.79 | 53.31 | 34.86 | 31.60 | 28.51 |
| Ours ($\lambda = 0.25$) | 26.61 | 25.94 | 31.73 | 73.39 | 56.77 | 44.48 |

from the test set to use as $x_c$ and another 2000 as $x_s$ to form 2000 content-style pairs. Next, we acquire the translated output $x_o$ and corresponding pseudo labels $y_c$, $y_s$, and $y_o$ by leveraging an off-the-shelf pretrained attribute classifier (EasyFace). In Table 4, we show the probabilities that the final generated image $x_o$ has an attribute from content image as $p(y_c^{att} = y_o^{att})$ and likewise for style image. Both ours and SAE are designed to make $z_s$ encode global high-level semantics, e.g., Gender, Age, etc. Thus, methods would show ideal performance if $y_o^{att} = y_s^{att} \neq y_c^{att}$. We see that most global attributes come from the content image for SAE indicating conservative translations from the style image (as seen in Fig. 4 and lower LPIPS in Table 1). In contrast, ours has a controllable way of deciding the strength of attributes from the style image through $\lambda$. The lower the value of $\lambda$, the more disentangled and consistent the attributes will be in the generations.

**Information Encoded in Each Latent Space.**
We analyze the role of the denoising network $\epsilon_\theta$ and the encoders $E_c$ and $E_s$ by analyzing what information is encoded in the respective latent spaces. Fig. 8 shows the results of fixing the content while varying the style images (and vice versa). $x_T$ is fixed as well to reduce the stochasticity. The remaining stochasticity comes from the white noise at each timestep during the reverse process. From the results, we can see that the structure information is maintained while style information changes according to the style image (and vice versa) as we intended. Similarly, in Fig. 17 in supplementary, we forward the same image to content and style encoders while the generation starts from different random noise $x_T$.

## 5 CONCLUSION

We propose a novel framework for enhancing controllability in image-conditioned diffusion models for reference-based image translation and image manipulation. Our content and style encoders trained along with the diffusion model do not require additional objectives or labels to learn to decompose style and content from images. The proposed generalized composable diffusion model extends CDM for a more generalized scenario. It shows significantly better performance when compared with CDM for translation as well as compositing text prompts. We also show that timestep-dependent weight schedules for conditioning inputs can help improve overall results and controllability. Additionally, the learned latent spaces are observed to have desirable properties like PCA-based attribute manipulation and smooth interpolations. Quantitative and qualitative evaluation shows the benefits of the proposed sampling techniques.

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
