# A Overview

Through this paper, we propose novel ways to condition and control diffusion models for image manipulation and translation. Our contributions can be summarized as follows:

- **Content and Style Latent Space:**
  We learn separate content and style latent spaces that correspond to different semantic factors of an image. This lets us control these factors separately to perform reference based image translation as well as controllable generation and image manipulation.

- **Timestep Scheduling:**
  We leverage the inductive bias of diffusion models and propose timestep dependent weight schedules to compose information from content and style latent codes for better translation.

- **Generalized Composable Diffusion Model (GCDM):**
  We extend Composable Diffusion Models (CDM) to allow for dependency between conditioning inputs, in our case, content and style. This results in significantly better generations and more controllability.

We believe that the proposed sampling techniques are applicable for controllable generation in general and not specific to image translation. To support our claims and show the ability of the proposed techniques for various problem formulations, we show additional results with various settings and parameters in appendix. We also provide more details on the experimental setup, models and parameters used to get the results shown in the main paper. Below is a list of contents in appendix.

1. We start by deriving how to decompose the score function of the joint conditioning to obtain the components in the proposed Generalized Composable Diffusion Model in Section B.

2. We next show a derivation that the GCDM PDF $\tilde{p}$ is proportional to a nested geometric average of different conditional distributions in Section C.

3. Preliminaries on Diffusion Models are provided in Section D.

4. We explain the training details, and show better identity preservation on faces and other datasets by combining the proposed sampling techniques with deterministic *reverse DDIM* (Preechakul et al., 2022) sampling in Section E.

5. We show that the learned style and content space can be used for attribute specific image manipulation in Section F. We also show that style and content interpolations between images can be used for for style/content transfer and mixing. We also analyze what the content and style encoders have learned by visualizing the nearest neighbors in content and style space respectively in Section F.3

6. The proposed timestep scheduling was only used during inference in the main paper. In Section G.1, we evaluate a model trained with timestep scheduling to implicitly learn a mixture–of–experts (Balaji et al., 2022) model by virtue of varying the conditional information at each timestep instead. We show these results on FFHQ in Fig. 13. We also show experiments with different timestep schedule functions in Section G.2

7. We finally provide some additional results such as text2image synthesis and reference based image translation in Section H.

# B DERIVATION FOR REFORMULATING THE SCORE FUNCTION OF THE JOINT CONDITIONING

One of the interesting properties of Diffusion Models is that a score function of the class conditional density $p(x|y)$ can be obtained by multiplying a score function of the marginal density $p(x)$ and a gradient of the likelihood $p(y|x)$. This was utilized in (Dhariwal & Nichol, 2021) for classifier guidance. Since classifier guidance requires a pre-trained classifier, (Ho & Salimans, 2022) proposed classifier-free guidance to control the generation.

## B.1 CLASSIFIER-FREE GUIDANCE (HO & SALIMANS, 2022)

$$\nabla_{x_t} \log p(x_t|c) = \nabla_{x_t} \log p(x_t, c) \tag{1}$$
$$\nabla_{x_t} \log p(x_t, c) = \nabla_{x_t} \log p(x_t)p(c|x_t) \tag{2}$$
$$= \nabla_{x_t} \log p(x_t)\frac{p(x_t|c)}{p(x_t)} \tag{3}$$
$$= \nabla_{x_t} \log p(x_t) + (\nabla_{x_t} \log p(x_t|c) - \nabla_{x_t} \log p(x_t)) \tag{4}$$
$$= \epsilon(x_t, t) + (\epsilon(x_t, t, c) - \epsilon(x_t, t)). \tag{5}$$

Practically, $\epsilon(x_t, t) + \alpha\left(\epsilon(x_t, t, c) - \epsilon(x_t, t)\right)$ is used where $\alpha$ is a temperature controlling the condition effect.

The $\nabla_{x_t} \log p(x_t|c) - \nabla_{x_t} \log p(x_t)$ can be seen as an implicit classifier.

## B.2 COMPOSABLE DIFFUSION MODELS (LIU ET AL., 2022)

$$\nabla_{x_t} \log p(x_t|c_1, c_2) = \nabla_{x_t} \log p(x_t, c_1, c_2) \tag{6}$$
$$\nabla_{x_t} \log p(x_t, c_1, c_2) = \nabla_{x_t} \log p(x_t)p(c_1, c_2|x_t) \quad assuming\ C_1 \perp C_2|X_t \tag{7}$$
$$= \nabla_{x_t} \log p(x_t)p(c_1|x_t)p(c_2|x_t) \tag{8}$$
$$= \nabla_{x_t} \log p(x_t)\frac{p(x_t|c_1)}{p(x_t)}\frac{p(x_t|c_2)}{p(x_t)} \tag{9}$$
$$= \nabla_{x_t} \log p(x_t) + \sum_i (\nabla_{x_t} \log p(x_t|c_i) - \nabla_{x_t} \log p(x_t)) \tag{10}$$
$$= \epsilon(x_t, t) + \sum_i (\epsilon(x_t, t, c_i) - \epsilon(x_t, t)) \tag{11}$$

Similar to the Classifier-free Guidance, hyperparameters for controlling the weight of each condition are used as well, i.e., $\epsilon(x_t, t) + \sum_i \alpha_i \left(\epsilon(x_t, t, c_i) - \epsilon(x_t, t)\right)$.

Now we introduce how to derive the components of GCDM formulation.

### B.3 GENERALIZED COMPOSABLE DIFFUSION MODELS

For brevity purposes, we omit the term that is canceled out because it is constant w.r.t. $x_t$, e.g., $\nabla_{x_t} \log p(c_1, c_2) = 0$ and $\nabla_{x_t} \log p(c_1) = 0$.

$$\nabla_{x_t} \log p(x_t|c_1, c_2) = \nabla_{x_t} \log p(x_t, c_1, c_2) \tag{12}$$

$$\nabla_{x_t} \log p(x_t, c_1, c_2) = \nabla_{x_t} \log p(x_t)p(c_1, c_2|x_t) \quad NOT \; assuming \; C_1 \perp C_2|X_t \tag{13}$$

$$= \nabla_{x_t} \log p(x_t)p(c_1|c_2, x_t)p(c_2|x_t) \tag{14}$$

$$= \nabla_{x_t} \log p(x_t)p(c_2|x_t) \left( \frac{p(c_2|c_1, x_t)p(c_1|x_t)}{p(c_2|x_t)} \right) \tag{15}$$

$$= \nabla_{x_t} \log p(x_t)p(c_2|x_t)p(c_1|x_t) \left( \frac{p(c_2|c_1, x_t)}{p(c_2|x_t)} \right) \tag{16}$$

$$= \nabla_{x_t} \log p(x_t)p(c_2|x_t)p(c_1|x_t) \left( \frac{p(c_1, c_2|x_t)}{p(c_1|x_t)p(c_2|x_t)} \right) \tag{17}$$

$$= \nabla_{x_t} \log \frac{p(x_t|c_2)p(x_t|c_1)}{p(x_t)} \left( \frac{\frac{p(x_t|c_1, c_2)}{p(x_t)}}{\frac{p(x_t|c_1)p(x_t|c_2)}{p(x_t)^2}} \right) \tag{18}$$

$$= \nabla_{x_t} \log \frac{p(x_t|c_2)p(x_t|c_1)}{p(x_t)} \left( \frac{p(x_t|c_1, c_2)p(x_t)}{p(x_t|c_1)p(x_t|c_2)} \right) \tag{19}$$

$$= \underbrace{-\epsilon(x_t, t) + \epsilon(x_t, t, c_2) + \epsilon(x_t, t, c_1)}_{\text{①}} \tag{20}$$

$$+ \underbrace{\{\epsilon(x_t, t, c_1, c_2) + \epsilon(x_t, t) - (\epsilon(x_t, t, c_1) + \epsilon(x_t, t, c_2))\}}_{\text{②}} \tag{21}$$

The term ② can be seen as a guidance from implicit classifiers.

$$\{\epsilon(x_t, t, c_1, c_2) + \epsilon(x_t, t) - (\epsilon(x_t, t, c_1) + \epsilon(x_t, t, c_2))\} \tag{22}$$

$$= \{\epsilon(x_t, t, c_1, c_2) - \epsilon(x_t, t) - (\epsilon(x_t, t, c_1) - \epsilon(x_t, t)) - (\epsilon(x_t, t, c_2) - \epsilon(x_t, t))\} \tag{23}$$

$$= \{\nabla_{x_t} \log p(x_t|c_1, c_2) - \nabla_{x_t} \log p(x_t) \tag{24}$$

$$- (\nabla_{x_t} \log p(x_t|c_1) - \nabla_{x_t} \log p(x_t)) - (\nabla_{x_t} \log p(x_t|c_2) - \nabla_{x_t} \log p(x_t))\} \tag{25}$$

$$= \{\nabla_{x_t} \log p(c_1, c_2|x_t) - (\nabla_{x_t} \log p(c_1|x_t) + \nabla_{x_t} \log p(c_2|x_t))\} \tag{26}$$

Similarly, the term ① can be rearranged as

$$- \epsilon(x_t, t) + \epsilon(x_t, t, c_2) + \epsilon(x_t, t, c_1) \tag{27}$$

$$= \epsilon(x_t, t) + (\epsilon(x_t, t, c_2) - \epsilon(x_t, t)) + (\epsilon(x_t, t, c_1) - \epsilon(x_t, t)) \tag{28}$$

$$= \nabla_{x_t} \log p(x_t) + \nabla_{x_t} \log p(c_2|x_t) + \nabla_{x_t} \log p(c_1|x_t) \tag{29}$$

By rearranging those two terms and adding hyperparameters $\alpha$, $\lambda$ and $\{\beta_c, \beta_s\}$, the proposed GCDM method in Definition 3.1 in the main paper can be obtained.

**Clarification of Eq. (17) and Eq. (18).** By Bayes rule, Eq. (17) becomes

$$\nabla_{x_t} \log \left[ \underbrace{p(x_t) \frac{p(x_t|c_2)p(c_2)}{p(x_t)} \frac{p(x_t|c_1)p(c_1)}{p(x_t)}}_{\text{①}} \underbrace{\left( \frac{\frac{p(x_t|c_1, c_2)p(c_1, c_2)}{p(x_t)}}{\frac{p(x_t|c_1)p(c_1)p(x_t|c_2)p(c_2)}{p(x_t)^2}} \right)}_{\text{②}} \right].$$

By rearranging ① and ② separately, the above equation becomes

$$= \nabla_{x_t} \log \left[ \underbrace{p(c_2)p(c_1)\frac{p(x_t|c_2)p(x_t|c_1)}{p(x_t)}}_{\text{rearranged from } \textcircled{1}} \underbrace{\left( \frac{\frac{p(x_t|c_1,c_2)}{p(x_t)}}{\frac{p(x_t|c_1)p(x_t|c_2)}{p(x_t)^2}} \right) \left( \frac{p(c_1,c_2)}{p(c_1)p(c_2)} \right)}_{\text{rearranged from } \textcircled{2}} \right].$$

By canceling out $p(c_1)p(c_2)$ in the first and the last term and by rearranging the equation, we can obtain Eq. (18), i.e.,

$$= \nabla_{x_t} \log \left[ \cancel{p(c_2)p(c_1)}\frac{p(x_t|c_2)p(x_t|c_1)}{p(x_t)} \left( \frac{\frac{p(x_t|c_1,c_2)}{p(x_t)}}{\frac{p(x_t|c_1)p(x_t|c_2)}{p(x_t)^2}} \right) \left( \frac{p(c_1,c_2)}{\cancel{p(c_1)p(c_2)}} \right) \right]$$

$$= \nabla_{x_t} \log \left[ \frac{p(x_t|c_2)p(x_t|c_1)}{p(x_t)} \left( \frac{\frac{p(x_t|c_1,c_2)}{p(x_t)}}{\frac{p(x_t|c_1)p(x_t|c_2)}{p(x_t)^2}} \right) p(c_1,c_2) \right]$$

$$= \underbrace{\nabla_{x_t} \log \left[ \frac{p(x_t|c_2)p(x_t|c_1)}{p(x_t)} \left( \frac{\frac{p(x_t|c_1,c_2)}{p(x_t)}}{\frac{p(x_t|c_1)p(x_t|c_2)}{p(x_t)^2}} \right) \right]}_{\text{Eq. (18)}} + \cancelto{0}{\nabla_{x_t} \log p(c_1,c_2)},$$

where $\nabla_{x_t} \log p(c_1,c_2) = 0$ because it is constant w.r.t. $x_t$.

## C   DERIVATION FOR COROLLARY 3.3.

The derivation starts from GCDM formulation proposed in Definition 3.1 in the main paper.

$$\nabla_{x_t} \log \tilde{p}_{\alpha,\lambda,\beta_1,\beta_2}(x_t|c_1,c_2) \triangleq \epsilon(x_t,t) + \alpha \Big[ \lambda(\underbrace{\epsilon(x_t,t,c_1,c_2) - \epsilon(x_t,t)}_{\nabla_{x_t} \log p(c_1,c_2|x_t)}) \tag{30}$$

$$+ (1-\lambda) \sum_{i=\{1,2\}} \beta_i \Big( \underbrace{\epsilon(x_t,t,c_i) - \epsilon(x_t,t)}_{\nabla_{x_t} \log p(c_i|x_t)} \Big) \Big].$$

Given the fact that $\epsilon(x_t,t) = \nabla_{x_t} \log p(x_t)$, taking integral w.r.t. $x_t$ to the equation yields:

$$\log \tilde{p}_{\alpha,\lambda,\beta_1,\beta_2}(x_t|c_1,c_2) = \log p(x_t) + \alpha \Big[ \lambda(\log p(x_t|c_1,c_2) - \log p(x_t)) \tag{31}$$

$$+ (1-\lambda) \sum_{i=\{1,2\}} \beta_i \Big( \log p(x_t|c_i) - \log p(x_t) \Big) \Big] + C,$$

where $C$ is a constant. Merging all the terms with log:

$$\log \tilde{p}_{\alpha,\lambda,\beta_1,\beta_2}(x_t|c_1,c_2) = \log \exp(C) + \log \Big( p(x_t) \left( \frac{p(x_t|c_1,c_2)}{p(x_t)} \right)^{\alpha\lambda} \left( \frac{p(x_t|c_1)^{\beta_1} p(x_t|c_2)^{\beta_2}}{p(x_t)^{\beta_1+\beta_2}} \right)^{\alpha(1-\lambda)} \Big)$$
$$\tag{32}$$

Taking exponential to the above equation:

$$\tilde{p}_{\alpha,\lambda,\beta_1,\beta_2}(x_t|c_1,c_2) = \exp(C)p(x_t)\left(\frac{p(x_t|c_1,c_2)}{p(x_t)}\right)^{\alpha\lambda}\left(\frac{p(x_t|c_1)^{\beta_1}p(x_t|c_2)^{\beta_2}}{p(x_t)^{\beta_1+\beta_2}}\right)^{\alpha(1-\lambda)} \tag{33}$$

$$= \exp(C)p(x_t)^{(1-\alpha\lambda-\alpha(1-\lambda)(\beta_1+\beta_2))}p(x_t|c_1,c_2)^{\alpha\lambda}\left(p(x_t|c_1)^{\beta_1}p(x_t|c_2)^{\beta_2}\right)^{\alpha(1-\lambda)} . \tag{34}$$

Given the fact that $\beta_1 + \beta_2 = 1$,

$$\tilde{p}_{\alpha,\lambda,\beta_1,\beta_2}(x_t|c_1,c_2) = \exp(C)p(x_t)^{(1-\alpha)}\left[p(x_t|c_1,c_2)^{\lambda}\left(p(x_t|c_1)^{\beta_1}p(x_t|c_2)^{(1-\beta_1)}\right)^{(1-\lambda)}\right]^{\alpha}, \tag{35}$$

Since the exponential function is always positive,

$$\tilde{p}_{\alpha,\lambda,\beta_1,\beta_2}(x_t|c_1,c_2) \propto p(x_t)^{(1-\alpha)}\left[p(x_t|c_1,c_2)^{\lambda}\left(p(x_t|c_1)^{\beta_1}p(x_t|c_2)^{(1-\beta_1)}\right)^{(1-\lambda)}\right]^{\alpha}, \tag{36}$$

which is the same as Eq. (7) in the main paper.

## D   PRELIMINARIES ON DIFFUSION MODELS

Diffusion Models (Sohl-Dickstein et al., 2015; Ho et al., 2020) are one class of generative models that map the complex real distribution to the simple known distribution. In high level, DMs aim to train the networks that learn to denoise a given noised image and a timestep $t$. The noised image is obtained by a fixed noising schedule. Diffusion Models (Sohl-Dickstein et al., 2015; Ho et al., 2020) are formulated as $p_\theta(x_0)$. The marginal $p_\theta(x_0)$ can be formulated as a marginalization of the joint $p_\theta(x_{0:T})$ over the variables $x_{1:T}$, where $x_1,...x_T$ are latent variables, and $p(x_T)$ is defined as standard gaussian. Variational bound of negative log likelihood of $p_\theta(x_0)$ can be computed by introducing the posterior distribution $q(x_{1:T}|x_0)$ with the joint $p_\theta(x_{0:T})$. In Diffusion Models (Sohl-Dickstein et al., 2015; Ho et al., 2020), the forward process $q(x_{1:T}|x_0)$ is a predefined Markov Chain involving gradual addition of noise sampled from standard Gaussian to an image. Hence, the forward process can be thought of as a fixed noise scheduler with the $t$-th factorized component $q(x_t|x_{t-1})$ represented as: $q(x_t|x_{t-1}) = \mathcal{N}(x_t; \sqrt{1-\beta_t}x_{t-1}, \beta_t I)$, where $\beta_t$ is defined manually. On the other hand, the reverse or the generative process $p_\theta(x_{0:T})$ is modelled as a denoising neural network trained to remove noise gradually at each step. The $t$-th factorized component $p_\theta(x_{t-1}|x_t)$ of the reverse process is then defined as, $\mathcal{N}(x_{t-1}|\mu_\theta(x_t,t), \Sigma(x_t,t))$. Assuming that variance is fixed, the objective of Diffusion Models (estimating $\mu$ and $\epsilon$) can be derived using the variational bound (Sohl-Dickstein et al., 2015; Ho et al., 2020) (Refer the original papers for further details).

Following Denoising Diffusion Probabilistic Models (Ho et al., 2020) (DDPM), Denoising Diffusion Implicit Model (Song et al., 2020) (DDIM) was proposed that significantly reduced the sampling time by deriving a non-Markovian diffusion process that generalizes DDPM. The latent space of DDPM and DDIM has the same capacity as the original image making it computationally expensive and memory intensive. Latent Diffusion Models (Rombach et al., 2022) (LDM) used a pretrained autoencoder (Esser et al., 2021) to reduce the dimension of images to a lower capacity space and trained a diffusion model on the latent space of the autoencoder, reducing time and memory complexity significantly without loss in quality.

All our experiments are based on LDM as the base diffusion model with DDIM for sampling. However the techniques are equivalently applicable to any diffusion model and sampling strategy.

## E   IMPLEMENTATION DETAILS

We build our models on top of LDM codebase[1]. For FFHQ and LSUN-church, we train our model for two days with eight V–100 GPUs. The model for AFHQ dataset is trained for one and a half days

---

[1] https://github.com/CompVis/latent-diffusion

with the same device. All models are trained for approximately 200000 iterations with a batch size of 32, 4 samples per GPU without gradient accumulation. All models are trained with $256\times256$ images with a latent $z$ size of $3\times64\times64$. The dimensions of content code $z_c$ is $1\times8\times8$ while that of style code $z_s$ is $512\times1\times1$. $t_1, t_2$ and $t_3$ from Eq. 1 in the main paper are timestep embeddings learned to specialize according to the latent code they are applied for to support learning different behavior for content and style features at different timesteps. We also experimented with different sizes for content and style code and chose these for best empirical performance. The content encoder takes as input $z$ and outputs $z_c$ following a sequence of ResNet blocks. The style encoder has a similar sequence of ResNet blocks followed by a final global average pooling layer to squish the spatial dimensions similar to the semantic encoder in (Preechakul et al., 2022).

To support GCDM during sampling, we require the model to be able to generate meaningful scores and model the style, content and joint distributions. Hence, during training we provide only style code, only content code and both style and content code all with probability 0.3 (adding up to 0.9) and no conditioning with probability 0.1 following classifier–free guidance literature. This helps learn the conditional and unconditional models that are required to use the proposed GCDM formulation. The code will be released upon acceptance of the paper.

During sampling, without *reverse DDIM*, if all the joint, conditionals, and unconditional guidance are used, sampling time for a single image is 10 seconds. With *reverse DDIM* to get $x_T$ where T is the final timestep, it takes 22 seconds. This might be lesser if *reverse DDIM* is stopped early and generation happens from the stopped point. Specific hyperparameters used to generate results in the main paper and appendix are provided in Table. 1.

Table 1: Hyperparameters used to generate the figures in the main paper and appendix. Timestep scheduling is only used in the sampling process. The parenthesis in the second column indicates the number of steps we used for sampling. Note that $\beta_c = 1 - \beta_s$.

| Dataset | sampler | $x_T$ | $\alpha$ | $\lambda$ | $\beta_s$ | $a$ | $b$ | scheduler |
|---|---|---|---|---|---|---|---|---|
| | | Main paper | | | | | | |
| FFHQ | DDIM+SDEdit (60) | *reverse DDIM* | 1.5 | 0.9 | 1.0 | 0.025 | 550 | sigmoid |
| LSUN-church | DDIM (100) | $q(x_{991}\vert x_0)$ | 2.0 | 0.5 | 0.0 | - | - | - |
| AFHQ | DDIM+SDEdit (60) | $q(x_{591}\vert x_0)$ | 3.0 | 0.75 | 1.0 | - | - | - |
| | | Appendix | | | | | | |
| FFHQ | DDIM (100) | *reverse DDIM* | 1.5 | 0.9 | 1.0 | 0.025 | 550 | sigmoid |
| LSUN-church | DDIM (100) | *reverse DDIM* | 5.0 | 0.5 | 0.0 | 0.025 | 600 | sigmoid |

We show results with various sets of parameters that can be used to control the effect of content, style and joint guidance in the generation in Fig. 1. The gray dotted box represents a baseline that we start from, and the rest of the other columns show the effects of each hyperparameter. As can be seen in the figure, each hyperparameter can be modified to get varying effects from style, content and joint guidance to get desirable results. We note that $\alpha$ works similar to classifier guidance scale in

|  | | Baseline | | | | | | | | | |
|---|---|---|---|---|---|---|---|---|---|---|---|
| Stronger style | $\alpha$ | 1.5 | 3.0 | 1.5 | 1.5 | 1.5 | 1.5 | 1.0 | 1.5 | 1.5 | 1.5 |
|  | $\lambda$ | 0.9 | 0.9 | 0.5 | 0.25 | 0.9 | 0.9 | 0.9 | 0.5 | 0.25 | 0.9 |
|  | $\beta_s$ | 1.0 | 1.0 | 1.0 | 1.0 | 1.0 | 1.0 | 1.0 | 0.0 | 0.0 | 1.0 |
| Stronger content | $a$ | 0.025 | 0.025 | 0.025 | 0.025 | 0.025 | 0.1 | 0.025 | 0.025 | 0.025 | 0.025 |
|  | $b$ | 550 | 550 | 550 | 550 | 650 | 550 | 550 | 550 | 550 | 450 |

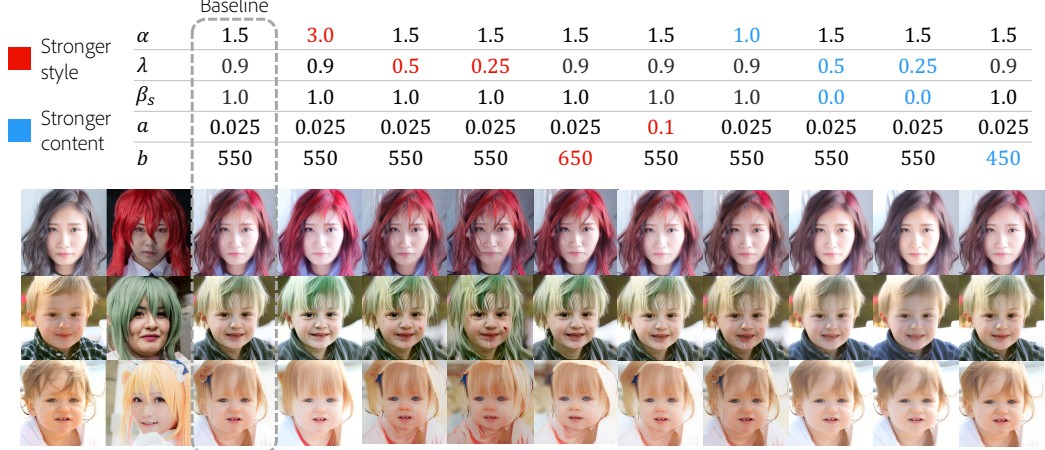

Figure 1: Example generations for different set of hyperparameters showing the effect of varying the controllable parameters during sampling. All generations start from $x_T$ obtained from reverse DDIM sampling using the content image.

diffusion models like (Rombach et al., 2022) and finding a good one when fixing $\lambda = 1$, $a = 0.025$, $b = 500$ takes lesser time (as independent content and style guidance is not provided when $\lambda = 1$). If the joint guidance has weaker style in the generations, it is recommended to modify $\lambda$ while setting $\beta_s = 1$. We find that this setting mostly gives good results. In rare cases when the style changes are limited even with smaller $\lambda$, increasing $b$ is another option. Modifying $a$ has relatively small effects on content and style specifically.

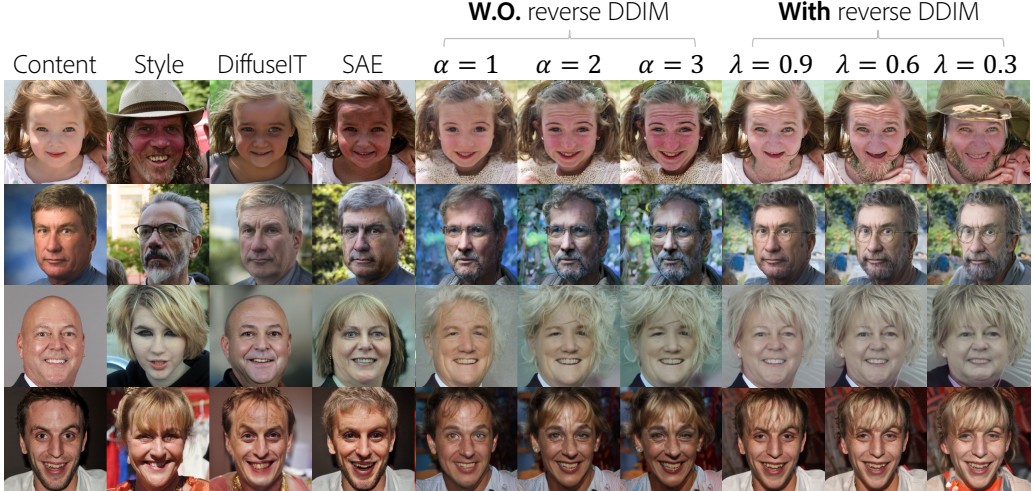

Figure 2: Comparisons between with and without *reverse DDIM* sampling for FFHQ model. We use the proposed GCDM and timestep scheduling during sampling. We can clearly see better identity preservation than not using reverse DDIM, on par with SAE (Park et al., 2020) while still providing better controllability.

Table 2: Quantitative comparison between the variants of ours with and without *reverse ddim*.

|  | w/o *reverse ddim* | | | w/ *reverse ddim* | | |
|---|---|---|---|---|---|---|
|  | Ours($\alpha = 1.0$) | Ours($\alpha = 2.0$) | Ours($\alpha = 3.0$) | Ours($\lambda = 0.9$) | Ours($\lambda = 0.6$) | Ours($\lambda = 0.3$) |
| FID | 20.38 | 23.68 | 26.45 | **11.99** | 13.40 | 15.45 |
| LPIPS | 0.53 | 0.57 | **0.6** | 0.34 | 0.42 | 0.49 |

### E.2 IDENTITY PRESERVATION

We notice that when the proposed sampling technique is used with randomly sampled noise $x_T$ for reference based image translation or manipulation, particularly on FFHQ dataset, that the identity of the content image is not preserved. This is an important aspect of image manipulation for faces. One of the ways to preserve better identity is to use the deterministic *reverse DDIM* process described in (Preechakul et al., 2022) to obtain $x_T$ that corresponds to a given content image. To do this, we pass the content image to both the content and style encoders as well as the diffusion model to get $x_T$ that reconstructs the content image. This $x_T$ is then used along with content code from content image and style code from style image to generate identity preserving translation.

The comparisons between with and without *reverse DDIM* during sampling are provided in Fig. 2. Columns 3-7 are the results reported in our main paper (Fig. 4), and Columns 8-10 are the results from using *reverse DDIM*. We can see that style is translated well on to the content image while preserving the identity of the content image. In contrast to SAE (Park et al., 2020) that preserves better identity by trading of magnitude of style applied, our approach provides the ability to control the magnitude of identity preservation and style transfer independently.

Additional comparisons between with and without *reverse DDIM* are provided in Fig. 3 and 4. We can see that the results with *reverse DDIM* better preserves the content identity while applying the style reasonably. On the other hand, the results without *reverse DDIM* have stronger impact of style with lesser identity preservation, which may be preferable in non-face domains such as abstract or artistic images or for semantic mixing.

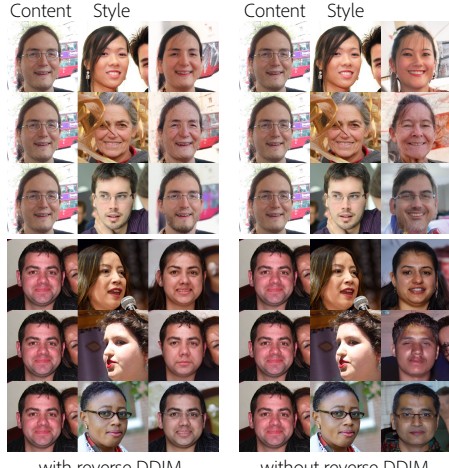

Figure 3: Comparisons between with and without DDIM reverse sampling method in FFHQ dataset.

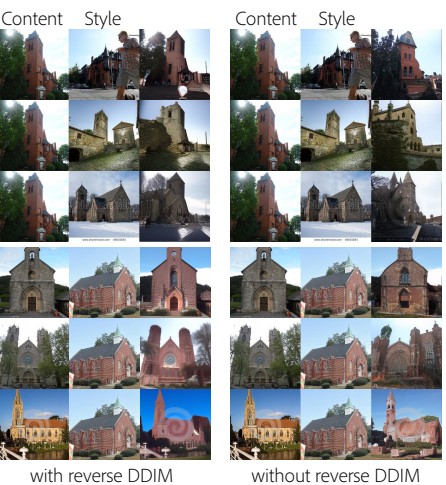

Figure 4: Comparisons between with and without DDIM reverse sampling method in LSUN-church dataset.

## F EXTENT OF CONTROLLABILITY

In this Section, we present rich controllability of our proposed framework. The latent space exploration is presented in Section F.1. Interpolation results are shown in Section F.2. Further analysis on the latent space is reported in Section F.3.

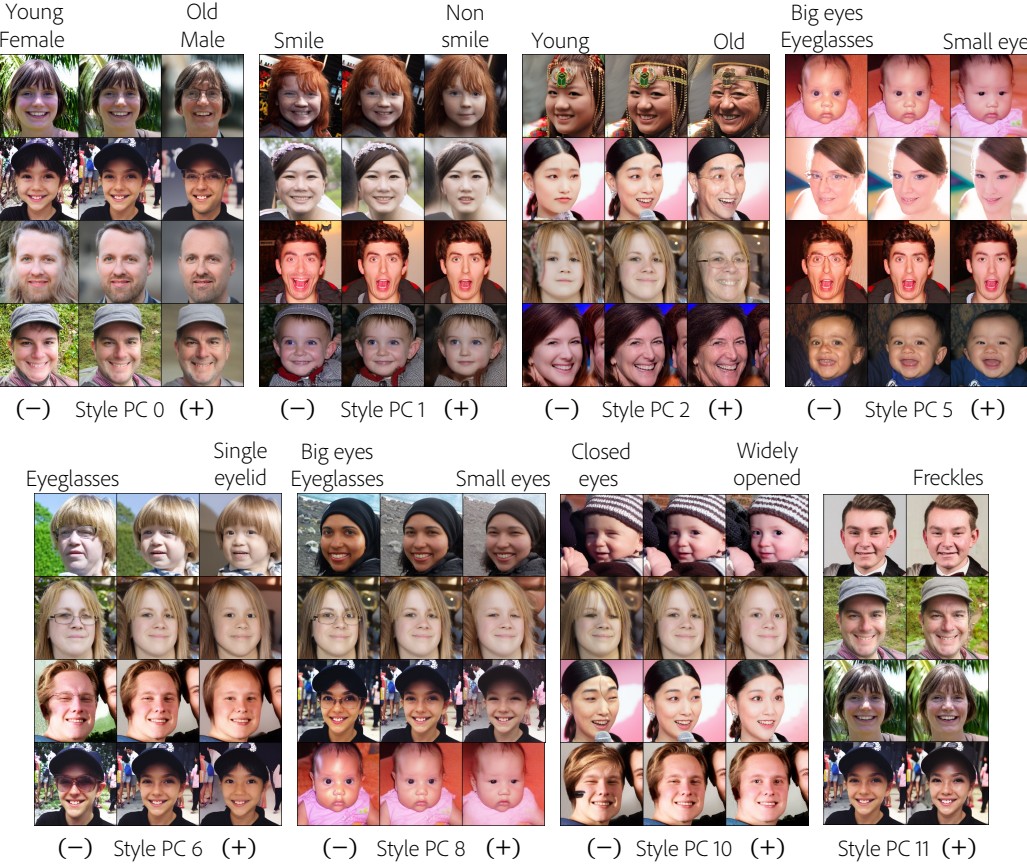

Figure 5: Example image manipulation results on FFHQ dataset by altering the style code learned by the proposed model. We can observe that each PCA component obtained from the style latent code control interpretable, meaningful semantic factors of an image such as *smiling*, *age*, *gender*, *hair color and texture*, *accessories* etc. The text above each block describes the attribute being modified while the bottom text refers to the principal component that causes the observed changes. Zoom in for better visibility.

## F.1 Image Manipulation with Latent Space Exploration

One of the advantages of the regularity of GAN latent space is the ability to find directions corresponding to specific attributes of an image that can be used for image manipulation. For example, (Härkönen et al., 2020) proposed to use the eigenvectors corresponding to the top principal components of the latent space to manipulate specific attributes. There are also other ways to find meaningful edit directions such as perturbing the dimensions corresponding to style vectors in StyleGAN (Wu et al., 2021) or using classifiers to find editable directions (Karras et al., 2020). DiffAE (Preechakul et al., 2022) also used classifiers to find editable directions in their semantic space.

To test the regularity and editability of the style and content latent spaces of the proposed model, we apply similar PCA (Härkönen et al., 2020) based techniques. We identify that the style codes control semantic information and the top principal components correspond to different attributes that can be seamlessly manipulated. We apply PCA algorithm on the style code $z_s$ and the content code $z_c$ of all the training images (60000) to get the top 30 Eigenvectors $\mathbf{V_{style}} = \{v_{style}^0, ..., v_{style}^{29}\}$ and $\mathbf{V_{content}} = \{v_{content}^0, ..., v_{content}^{29}\}$. The obtained basis vectors are used for shifting each individual sample.

To manipulate the attribute controlled by the style principal component (PC) on an image $\mathbf{I}$, we pass the content code $z_c^I$, style code of the same image modified as $z_s'^I = z_s^I + w_s \cdot v_{style}^0$ and the noise $x_T$ obtained by applying *reverse DDIM* with content image, for generation. Similar procedure is

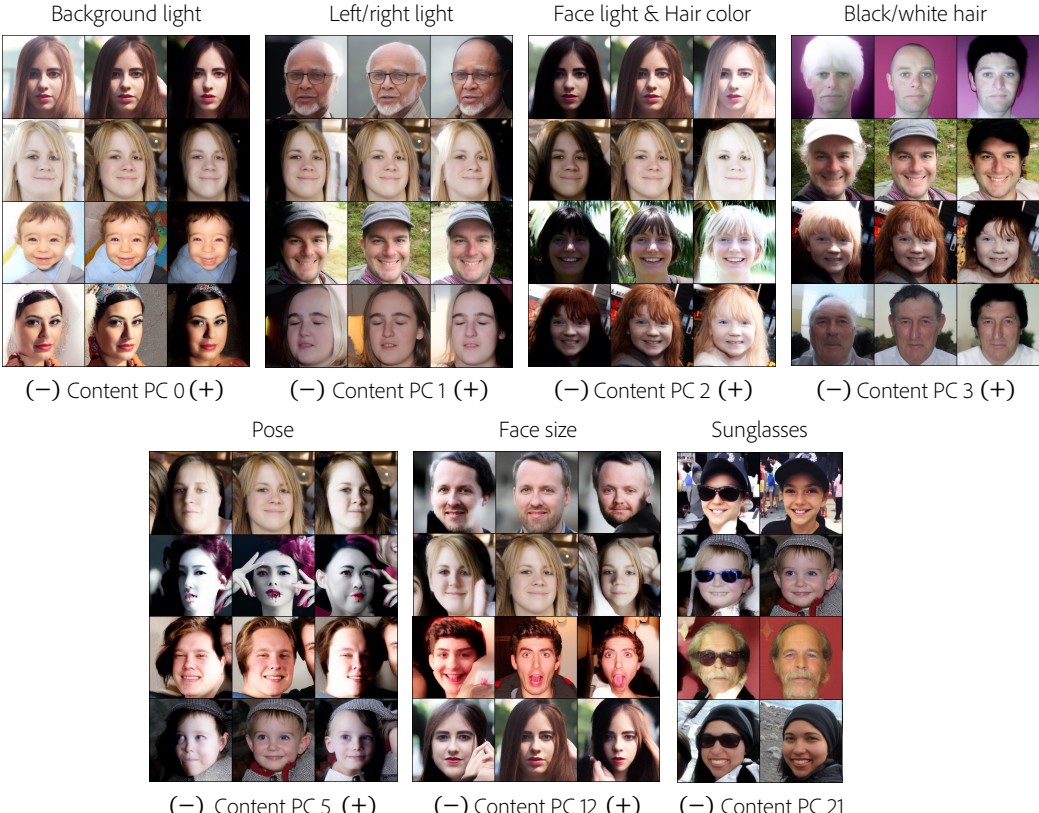

Figure 6: Example image manipulation results on FFHQ dataset by manipulating the content code learned by the proposed model. The components from content code shows controls that correspond to spatial attributes such as pose, lighting and background. The text above each block describes the attribute being modified while the bottom text refers to the principal component that causes the observed changes. Zoom in for better visibility.

followed for the content code as well. Please note that PCs are obtained from the training samples while unseen data is taken as input **I** for the PC experiments.

We performed the analysis with FFHQ and LSUN-church dataset. In FFHQ results, as shown in Fig.5, we observe that the PCs of the style space contain meaningful high-level semantics. For example, the first PC $v_{style}^0$ controls *gender* and *age*. This indicates that the style encoder learns as intended under our proposed framework. Example generations for $w_s \in \{-3, 0, 3\}$ are shown for various PCs. Fig. 6 shows the results of manipulating the content codes in FFHQ dataset. Interestingly, the content PCs encode the spatial-relevant information, such as the light in the background, light on the left and right, pose, and facial shape. For the content PC experiments, $w_c \in \{-1, 0, 1\}$ is used.

We also explored the same experiments with LSUN-church dataset. Since the foreground region of LSUN-church is not as simple and consistent as that of FFHQ dataset, the content PC results are not consistent. However, we could find some meaningful style PCs because it is designed to contain global features. As seen in Fig. 7 design, texture, abstraction, color are some attributes that are controllable. Images are obtained with $w_s \in \{-2, 0, 2\}$. We believe that using classifiers can possibly lead to better directions for manipulation but it is interesting that simpler PCA based technique provide meaningful semantic edit directions.

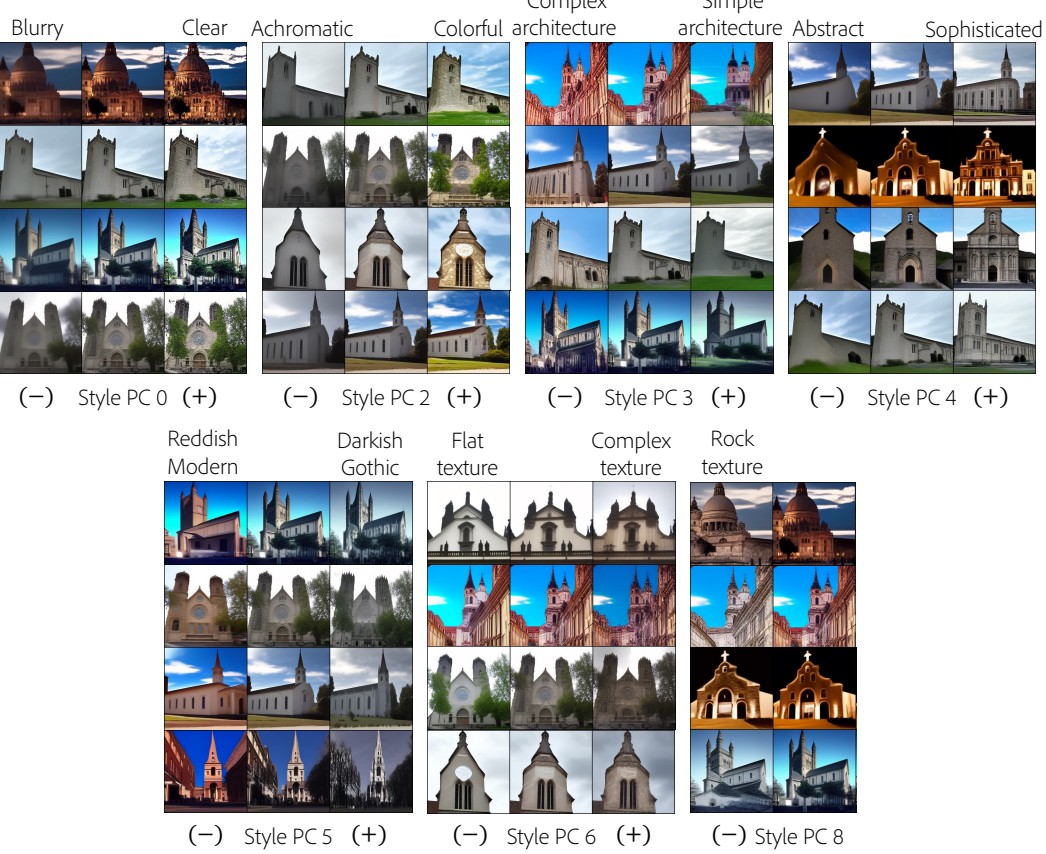

Figure 7: Example image manipulation results on LSUN church dataset by altering the style code learned by the proposed model. We can observe that each PCA component obtained from the style latent code control interpretable, meaningful semantic factors of an image such as *texture*, *color*, *abstraction* and *design*. The text above each block describes the attribute being modified while the bottom text refers to the principal component that causes the observed changes. Zoom in for better visibility.

## F.2 INTERPOLATION

We conducted experiments on the latent space interpolation in order to analyze the effects of the content, the style, and $x_T$ during the sampling process. All the results use reverse DDIM with content image to get $x_T$ that is used during sampling.

Fig. 8 shows the content-only interpolation results where style code $z_s$ and noise $x_T$ are fixed to the image in the first column. The gray box on the top indicates the fixed input while $z_c$ is interpolated between the two images in the first two columns. From the figure, we can see that the style information and the person identity are maintained while pose and facial shape are changed.

Fig. 9 shows the case $x_T$ obtained from reverse DDIM of the images in the first two columns is interpolated while the style and content features are fixed to the image in the first column. The content (e.g., pose, facial shape) and the style (e.g., beard, eyeglasses, and facial color) are maintained while stochastic properties change. We can see that identity is not entirely tied to $x_T$ but the stochastic changes to cause change in the identity. This is why using reverse DDIM to fix $x_T$ preserves better identity.

Fig. 10 visualizes the style interpolation while content and $x_t$ are fixed to the first image. The person identity, the pose and facial shape are preserved while the facial expression, gender, and age are smoothly changed validating our results.

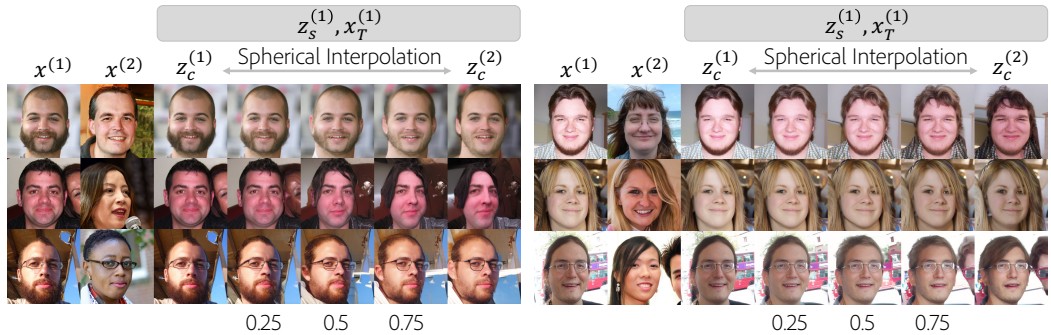

Figure 8: Content interpolation results. Style and $x_T$ are obtained from images in the first column while content code is interpolated between images in column 1 and column 2. We can see how content specific factors vary smoothly.

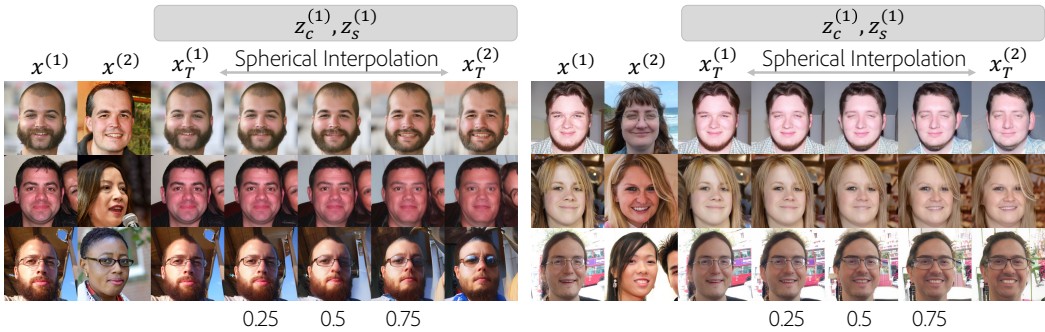

Figure 9: $x_T$ interpolation results where style and content are obtained from images in the first column while $x_T$ is interpolated between reverse DDIM of both images. We can see stochastic changes causing mild identity changes. Fixing $x_T$ to the content image hence provides better identity preservation for image translation and manipulation.

## F.3 INTERPRETING THE LATENT SPACES

We additionally perform K Nearest Neighbor (KNN) experiments to understand what features are encoded in the content and style latent representations. We pass 10000 unseen images through the

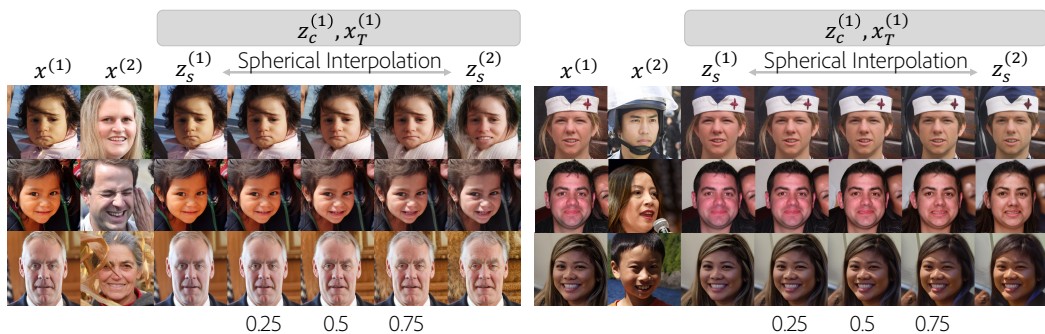

Figure 10: Style interpolation results when content and $x_T$ are obtained from images in the first column. We can see smooth changes in the semantic attributes such as age, gender, smile, eyeglasses etc. allowing for effective style manipulations.

style and the content encodersto get $z_c$ and $z_s$. We then compute the distance of an arbitrary sample with the entire validation set and sort the 10000 distances.

The results are shown in Fig. 11 and Fig. 12. The first column denotes the input image while the rest of the columns shows the top 10 images that have the closest content or style features indicated by $z_c$ (first row within each macro row) and $z_s$ (second row within each macro row) respectively. The second column is the same image. We can see that the content feature mainly contains the pose and the facial shape information while the style has the high-level semantics, such as wearing eyeglasses, gender, age, accessories, and hair color.

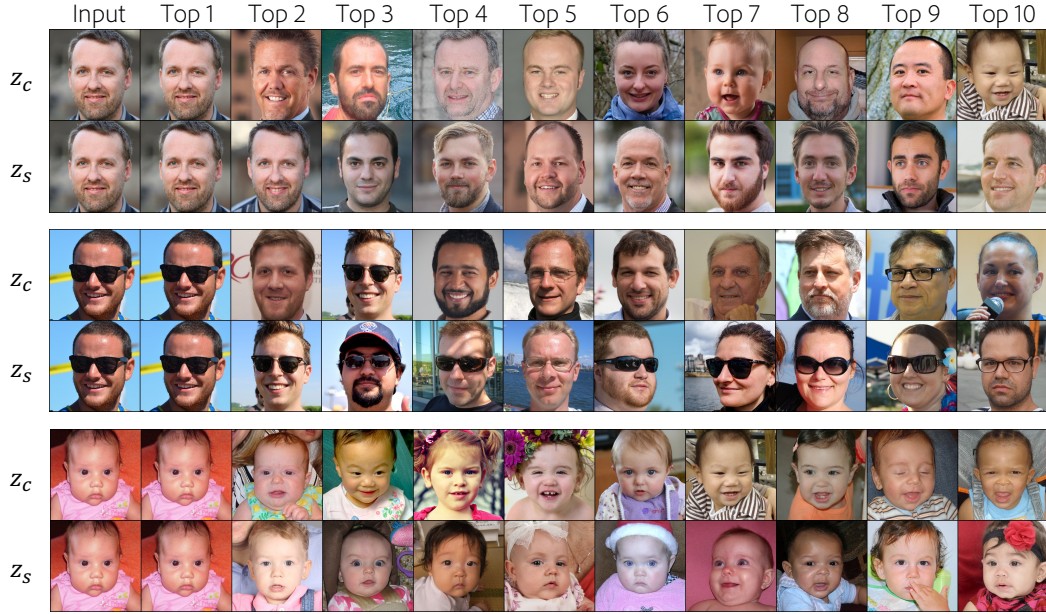

Figure 11: KNN results of the content and the style features showing what semantic attributes content and style codes encode.

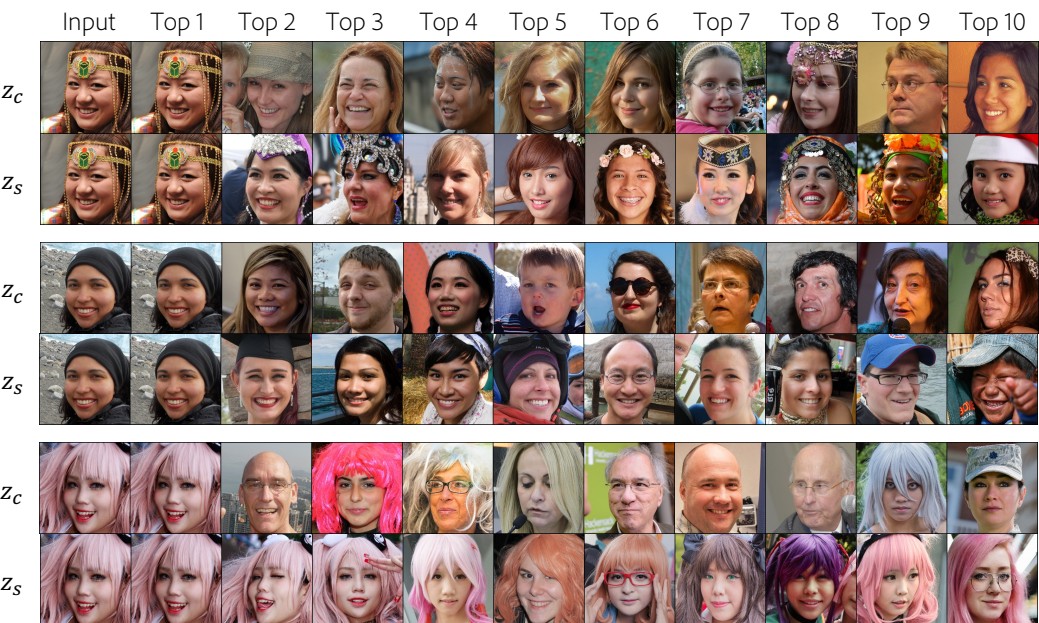

Figure 12: KNN results of the content and the style features showing what semantic attributes content and style codes encode.

# G    TIMESTEP SCHEDULING

## G.1    TRAINING AN IMPLICIT MIXTURE-OF-EXPERTS

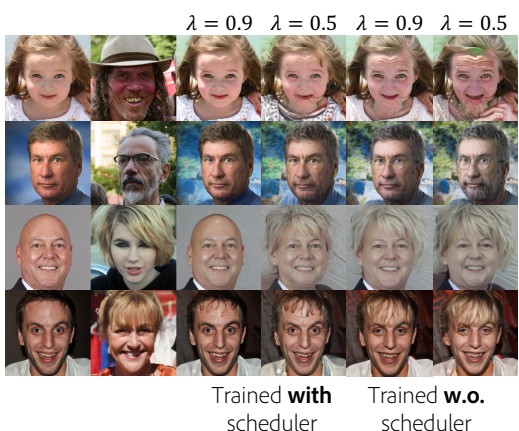

Figure 13: Effects of using the proposed timestep scheduling in training.

Our timestep scheduling approach proposed in Sec 3.2 in the main paper was applied only during sampling for results in the main paper. We trained a model with timestep scheduling applied during training to analyze how it affects the behavior of our framework. Fig. 13 shows the comparisons between the models trained with and without the scheduler. For the results trained with scheduler, we used $a = 0.1$ and $b = 529$ ($\text{SNR}^{-1}(0.1)$) for both training and sampling. As can be seen in the rightmost two columns, the style effects are relatively small although given $\lambda$ is controlled. It is because the style encoder is trained to be injected only in the early timesteps (0-528), which makes the style representations learn limited features (e.g., eyeglasses are not encoded in the style, as shown in the second row). However, we observe better decomposition between factors controlled by content and style compared to using the timestep scheduling only during sampling. We believe this is because, using timestep scheduling to vary the conditioning input at each timestep implicitly trains the model to specialize to the varied conditioning, implictly learing a mixture–of–experts like model (Balaji et al., 2022). We believe this could be a promising avenue for future research to train expert models without maintaining different entirely finetuned models and leave further analysis as future work.

## G.2    EXPERIMENT WITH DIFFERENT TIMESTEP SCHEDULES

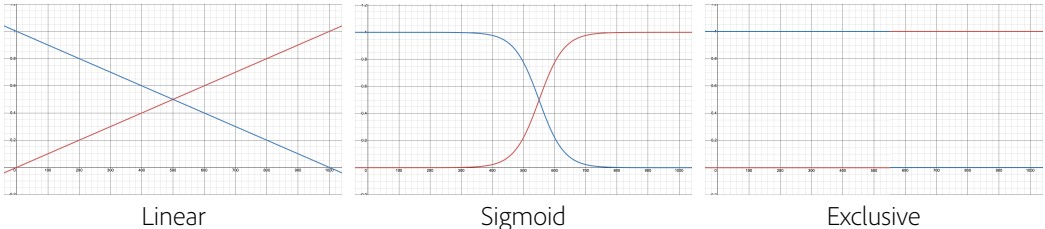

Figure 14: Plots for different timestep scheduling strategies. The illustrated plot of the sigmoid scheduler is from $a = 0.025$ and $b = 550$. Bigger $a$ makes it similar to the exclusive scheduler while smaller $a$ makes it close to the linear scheduler. The blue line indicates the weight scheduler for the style and the red line is for the content.

We compare the different timestep schedulers illustrated in Fig. 14 during sampling. Note that these schedules are not used for training. In the exclusive scheduling, the style weight is one if $t \leq 550$ and zero otherwise. The content weight is applied when style weight is not applied. In the linear

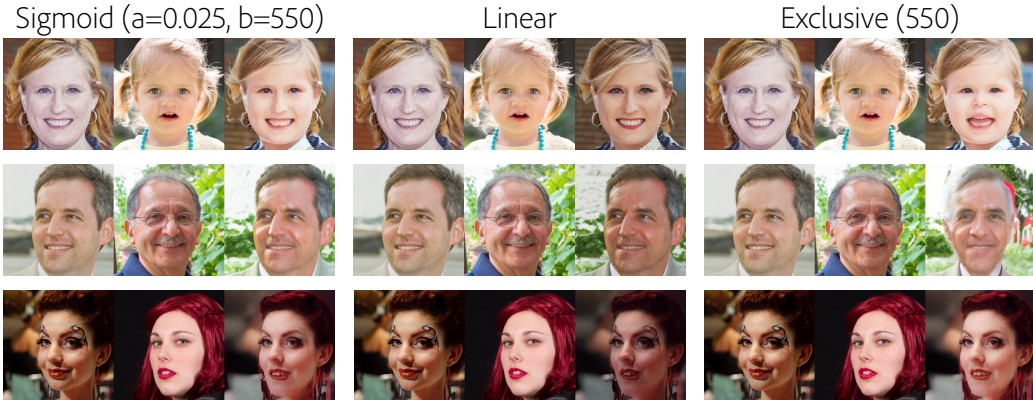

Figure 15: Comparison between different timestep scheduled during sampling. Sigmoid has a softer schedule with more controllability and thus results is more natural generations compared to the other techniques.

scheduling, the style weight linearly decreases from 1 at $t = 0$ to 0 at $t = 999$ while the content weight increases linearly from 0 to 1. The sigmoid scheduling is the one propose in Eq. 2 and 3.

The comparison results are shown in Fig. 15. We can observe that the exclusive scheduling shows either magnified style or unnatural generations compared to the sigmoid scheduling. Since it is difficult to exactly define the role of each timestep, naively separating the point where to exclusively apply content and style yields the undesirable results. The linear schedule does not work for all images and has limited control. However, the sigmoid scheduling provides a softer weighting scheme leading to better generations and has additional controls to get desired results.

# H  ADDITIONAL RESULTS

In this Section, we provide additional results of our proposed framework. Fig. 16 shows example generations using CDM and GCDM from the same model. CDM consistently shows two failure modes specific to reference based image translation. First, content-style overlap showing that content and style codes have certain common information that is not further disentangled during sampling (e.g., first and second rows in the figure). Next is unnatural generation (e.g., third and fourth rows) where the generated images do not look realistic enough.

Fig. 17 shows the effect of the starting point $x_T$ given same content and style codes. Fig. 18 shows the additional results on FFHQ dataset. Fig. 19 shows the results of multiple style and single content, and vice versa on LSUN church dataset. Fig. 20 shows the hyperparameters used for CDM and GCDM on Stable Diffusion V2.

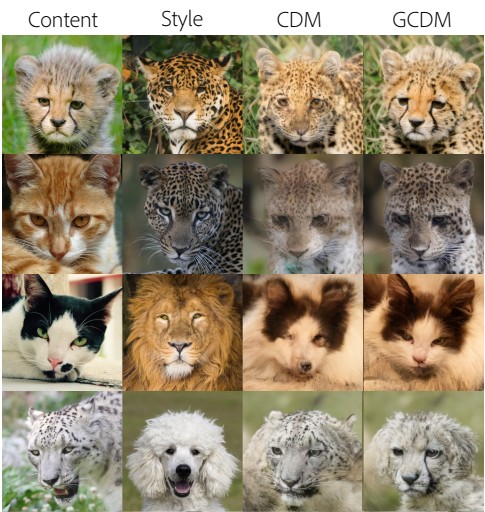

Content    Style    CDM    GCDM

Figure 16: Comparisons between GCDM and CDM demonstrating that CDM can output unnatural images while GCDM can generate realistic images. We use DDIM (Song et al., 2020) sampler, and the reverse process is done from $T = 600$ inspired by SDEdit (Meng et al., 2021). $z_{600}$ is obtained by $q(z_{600}|E_{LDM}(x_c))$ using the content image. $x_T$ is randomly sampled.

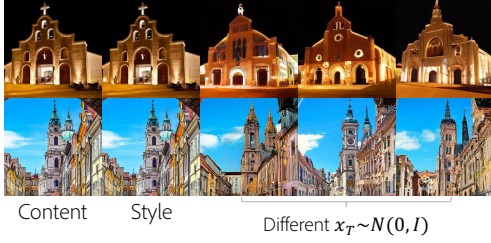

Content    Style    Different $x_T \sim N(0, I)$

Figure 17: Example showing the role of the denoising network during sampling when content and style codes are unchanged. $x_T$ is randomly sampled. The images show that the denoising network play a role in stochasticity since the outputs have consistent shape, color and texture information while minor details of the buildings or clouds are changed.

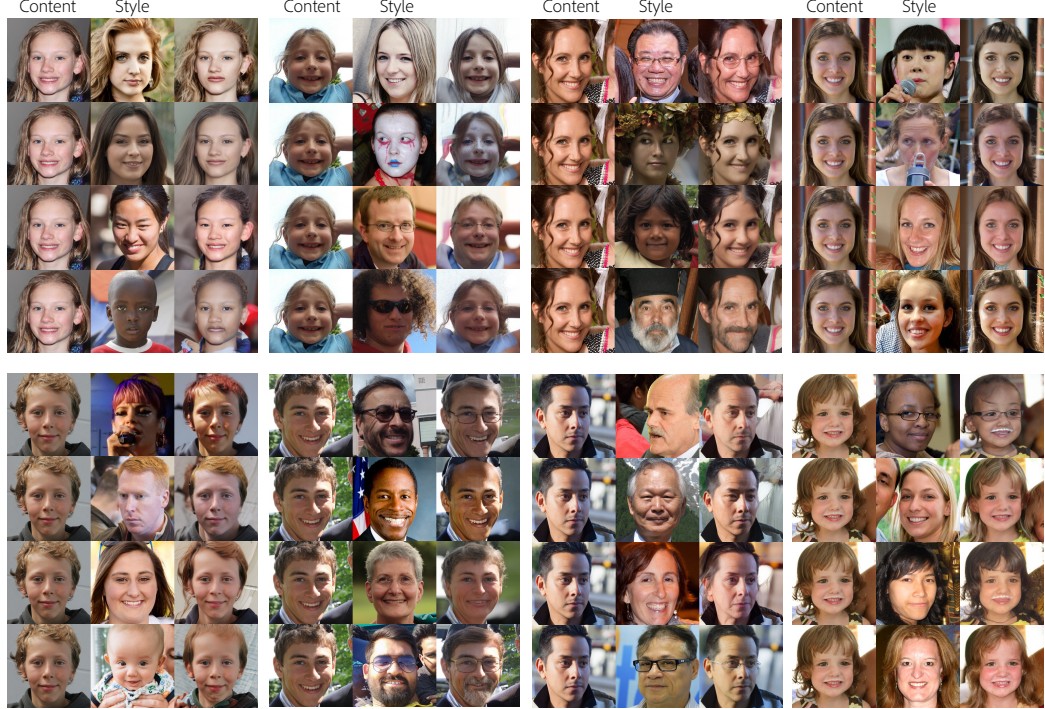

Figure 18: Additional results on FFHQ. The results are sampled by *reverse DDIM*.

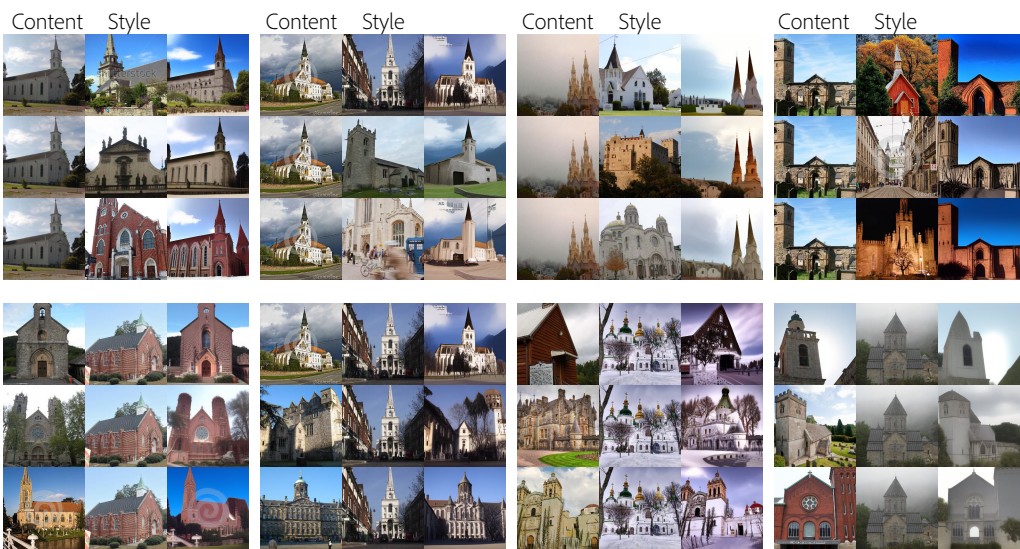

Figure 19: Additional results on LSUN-church. The results are sampled by *reverse DDIM*.

$c_1$ = Photo of a bear
$c_2$ = Photo of a car in the red forest
$c_{1,2}$ = Photo of a bear and a car in the red forest

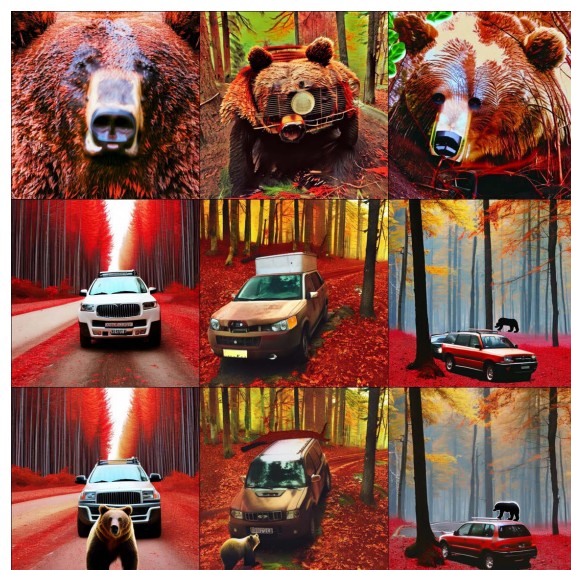

CDM
$(\alpha = 9.0, \lambda = 0.0, \beta_1 = 1.0, \beta_2 = 1.0)$

GCDM
$(\alpha = 9.0, \lambda = 1.0)$

GCDM
$(\alpha = 9.0, \lambda = 0.85, \beta_1 = 1.0, \beta_2 = 0.0)$

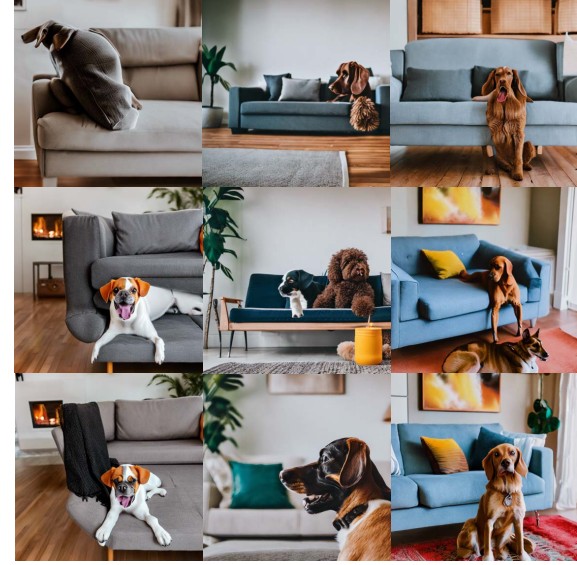

CDM
$(\alpha = 9.0, \lambda = 0.0, \beta_1 = 1.0, \beta_2 = 1.0)$

GCDM
$(\alpha = 9.0, \lambda = 1.0)$

GCDM
$(\alpha = 9.0, \lambda = 0.5, \beta_1 = 0.0, \beta_2 = 1.0)$

$c_1$ = Photo of a couch
$c_2$ = Photo of a dog sitting in the living room
$c_{1,2}$ = Photo of a couch and a dog sitting in the living room

Figure 20: Text2image synthesis results with GCDM hyperparameters.