# OpenReview forum: "Towards Enhanced Controllability of Diffusion Models"
_ICLR.cc/2024/Conference — Submitted to ICLR 2024_

### Official Review · Reviewer_e2QG · 2023-10-31

**Soundness:** 3 good
**Presentation:** 3 good
**Contribution:** 2 fair
**Rating:** 3
**Confidence:** 5

**Summary:**

This work addresses the challenge of improving the controllability of Diffusion Models, which have shown significant capabilities in image generation. The authors introduce three methods to enhance the controllability of these models during both training and sampling phases. They train Diffusion Models conditioned on two latent codes: a spatial content mask and a flattened style embedding. The paper also introduces two sampling techniques to improve controllability and demonstrates that their methods allow for effective image manipulation and translation.

**Strengths:**

- The paper addresses a significant gap in the controllability of Diffusion Models, especially in feature disentanglement and composing multiple conditions.
- The method achieves a good performance on FFHQ, AFHQ, LSUN datasets.

**Weaknesses:**

- So called style condition looks more like another object feature condition. The style image thus is quite close to the content image on the manifold.
- All examples of content and style pairs are quite similar, i.e. both faces, cat and dog, cat and tiger. This undermines the effectiveness of this method on more general cases.

**Questions:**

- Can you explain how do you apply your method to SD text-to-image generation in detail?
- Have you tried using another style image that is more different in style instead of just content? It seems more like a object feature transfer case in your experiments. Can you try an obviously different style image, such as a painting etc?

**Details Of Ethics Concerns:**

None.

---

> ### Author Response · Authors · 2023-11-22
> **Response to Reviewer e2QG**
>
> > (Q1) Can you explain how do you apply your method to SD text-to-image generation in detail?
>
> given a caption of "photo of a red bird and a yellow flower", we need to compute three terms for obtaining $\tilde{p}$, the GCDM formulation (Eq. 3 in the main paper). In specific, $\epsilon(x_t, t, c_1)$ is obtained by forwarding "photo of a red bird" and $\epsilon(x_t, t, c_2)$ is obtained by forwarding "photo of a yellow flower". Lastly, the joint term $\epsilon(x_t, t, c_1, c_2)$ is obtained by forwarding "photo of a red bird and a yellow flower".
>
> > (W2) All examples of content and style pairs are quite similar, i.e. both faces, cat and dog, cat and tiger. This undermines the effectiveness of this method on more general cases.
>
> FFHQ, AFHQ, and LSUN-church datasets are widely-used benchmark datasets to evaluate the performance of generative models in Image Translation tasks [1-4].
>
> [1] Multimodal Unsupervised Image-to-Image Translation, Huang et al., ECCV'18
>
> [2] StarGAN v2: Diverse Image Synthesis for Multiple Domains, Choi et al., CVPR'20
>
> [3] Swapping Autoencoder for Deep Image Manipulation, Park et al., NeurIPS'20
>
> [4] Diffusion Autoencoders: Toward a Meaningful and Decodable Representation, Preechakul et al., CVPR'22
>
> > (W1) So called style condition looks more like another object feature condition. The style image thus is quite close to the content image on the manifold.
>
> > (Q2)  Have you tried using another style image that is more different in style instead of just content? It seems more like a object feature transfer case in your experiments. Can you try an obviously different style image, such as a painting etc?
>
> As shown in Fig. 7 in the main paper and Fig. 18-19 in the supplementary, our style encoder encodes non-structural information, e.g., textual information (in LSUN-church and AFHQ) and facial attribute information (in FFHQ) while structural information is encoded in the content encoder. (Please see Table 4 if the reviewer wants to see the quantitative verification.) Similarly, we strongly believe that the low-frequency texture information would be encoded in our style encoder if we train our framework with (only) Art dataset. This is because the non-structural information in Art dataset mainly includes the low-frequency texture of the arts, and the style encoder is designed to learn the non-structural information as verified by the aforementioned experiments.

---

### Official Review · Reviewer_gF5P · 2023-10-31

**Soundness:** 2 fair
**Presentation:** 3 good
**Contribution:** 2 fair
**Rating:** 3
**Confidence:** 4

**Summary:**

This paper proposes to enhance the controllability of diffusion models by adding a content encoder and a style encoder in the diffusion models. Using the same design philosophy as Swapping Autoencoders, the content encoder learns a spatial layout mask while the style encoder outputs the flattened semantic codes. By doing so, the method enables deep image manipulation by mixing the style codes and the content codes. Furthermore, the paper proposes a timestep-dependent weighting schedule for content and style latents to get better results. Experiments are mostly done on FFHQ datasets as well as LSUN-Church and AFHQ.

**Strengths:**

1. The method is simple and easy to understand.
2. The controllability of diffusion models is an important problem and worth investigating.

**Weaknesses:**

1. The method adds two additional encoders and needs to be trained from scratch which makes it hard for larger text-to-image diffusion models which require a large amount of time training.
2. Swapping autoencoders is a strong baseline here and it does not seem the proposed model shows significant advantages over this previous method from Figure 4. For example, the results from SAE seem more plausible than the different results given by the proposed method.
3. From Figure 15 on the ablation of different weighting schedules, the difference seems little between the sigmoid and the linear ones.

**Questions:**

For the comparison with DiffAE+MagicMix, it seems that the image quality is really bad, does the author try different parameters over this? The paper states that the model takes $x_600$ as input but no other results are presented here.

---

> ### Author Response · Authors · 2023-11-22
> **Response to Reviewer gF5P**
>
> > (W1) The method adds two additional encoders and needs to be trained from scratch which makes it hard for larger text-to-image diffusion models which require a large amount of time training.
>
> We believe the two additional encoders can be trained by fine-tuning the pre-trained Diffusion Models (not necessarily training from scratch). FastComposer [1] and ControlNet [2] have shown that it is possible to train their additional encoders while fine-tuning or fixing the off-the-shelf pretrained Diffusion Models. Given the recent research reports in this field, it should be possible to train the additional encoders in a fine-tuning manner.
>
> [1] FastComposer: Tuning-Free Multi-Subject Image Generation with Localized Attention, Xiao et al., ArXiv'23
>
> [2] Adding Conditional Control to Text-to-Image Diffusion Models, Zhang et al., ICCV'23
>
> > (W2) Swapping autoencoders is a strong baseline here and it does not seem the proposed model shows significant advantages over this previous method from Figure 4. For example, the results from SAE seem more plausible than the different results given by the proposed method.
>
> SAE might show more plausible results in some cases, but our results $\textbf{generally}$ show better performance than the baseline in FID and LPIPS (See Table 1 and Table 2). Furthermore, we have more controllability than SAE about how much style will be applied and how much content information will be preserved. The extended controllability can be easily understood by seeing Fig. 1 in the supplementary. Also, additional results can be found in Fig. 18 and Fig. 19 in the supplementary.
>
> > (W3) From Figure 15 on the ablation of different weighting schedules, the difference seems little between the sigmoid and the linear ones.
>
> thank you for pointing out a valid point. Even though the performance difference is not huge, we can have additional extended control if we use the sigmoid scheduling. Please see Fig. 1 in the supplementary. The results on the fifth and sixth columns from the right (compared to the baseline on the third column from the left) show that we can have additional control, which is not achievable in the linear scheduling. Please note that both the sigmoid and the linear scheduling are proposed by us.
>
> > (Q1) For the comparison with DiffAE+MagicMix, it seems that the image quality is really bad, does the author try different parameters over this? The paper states that the model takes
>  as input but no other results are presented here.
>
> For this experiment, we used $K_{max}=0.6T$ and $K_{min}=0.3T$ with $v=0.5$, which are suggested by MagicMix [3] as the best hyperparameters. Please see "Preserving more layout details" paragraph on page 8 of their paper.
>
> [3] MagicMix: Semantic Mixing with Diffusion Models, Liew et al., ArXiv'22

---

### Official Review · Reviewer_4fTL · 2023-11-01

**Soundness:** 3 good
**Presentation:** 3 good
**Contribution:** 2 fair
**Rating:** 6
**Confidence:** 4

**Summary:**

Generative models have become increasingly popular in recent years, but there is still a need to improve their controllability. Diffusion models are a promising class of generative models, but existing methods for disentangling latent spaces in diffusion models do not effectively learn multiple controllable latent spaces. This paper proposes a novel framework for enhancing the controllability of diffusion models. The framework introduces a Content Encoder and a Style Encoder to better manage the structure and style aspects of an image during training. Additionally, it presents Generalized Composable Diffusion Models (GCDM) to allow for more natural compositions during inference when conditional inputs are not independent.  The paper also utilizes the inductive bias of diffusion models to improve results by applying a controllable timestep-dependent weight schedule to blend content and style codes during generation.

**Strengths:**

1. Enhanced Controllability: The proposed framework introduces a novel approach to enhance controllability in generative models, specifically Diffusion Models, by effectively learning two latent spaces—content and style. This level of control is crucial for a wide range of practical applications in image synthesis and beyond.

2. Disentanglement of Latent Spaces: The framework addresses a gap in existing research by effectively disentangling latent spaces. The introduction of separate Content and Style Encoders helps in managing the structural and stylistic aspects of an image more precisely.

**Weaknesses:**

1. Complexity of Implementation: The introduction of multiple encoders and the management of separate latent spaces could increase the complexity of the model's architecture, making it more challenging to implement and fine-tune.

2. Overlapping latent spaces: Are the latent spaces independent of each other?  How can this be verified?

3. Justification of success can be strengthened: Are there specific domains or types of images where the proposed method performs particularly well or poorly? What are the limitations in terms of content and style diversity?

**Questions:**

1. Could the authors provide quantitative metrics to compare the performance of their proposed GCDM with existing models such as CDM and others?
2. Is there a possibility to extend the framework to more than two latent spaces, and if so, how would this affect the model's complexity and performance?
3. How does the proposed timestep-dependent weight schedule compare with existing methods in terms of computational efficiency and quality of generated images?
4. The formulation of the style coder is based on human heuristic, instead of data driven.  How can you tell what styles can be controlled?

---

> ### Author Response · Authors · 2023-11-22
> **Response to Reviewer 4fTL**
>
> > (W1) Complexity of Implementation: The introduction of multiple encoders and the management of separate latent spaces could increase the complexity of the model's architecture, making it more challenging to implement and fine-tune.
>
> Thank you for pointing out a valid point. We agree with the fact that the model architecture becomes more complex by introducing two encoders. However, we would like to emphasize that we can have more controllability than having a single encoder.
>
> > (Q2) Is there a possibility to extend the framework to more than two latent spaces, and if so, how would this affect the model's complexity and performance?
>
> Similar to the previous studies [1,2], our content encoder mainly encodes spatial information and the style encoder mainly encodes non-spatial information. And it is done by leveraging the inductive bias of Diffusion Models under the sophisticated design of our proposed framework. Thus, it is not possible to add the third encoder under the current framework. However, with our proposed two spaces, we can have extended controllability such as finding semantically meaningful direction vectors (Fig. 5-7 in the supplementary) and conducting interpolation in the latent spaces (Fig. 8-10 in the supplementary)
>
> [1] Multimodal Unsupervised Image-to-Image Translation, Huang et al., ECCV'18
>
> [2] Swapping Autoencoder for Deep Image Manipulation, Park et al., NeurIPS'20
>
> > (W2) Overlapping latent spaces: Are the latent spaces independent of each other? How can this be verified?
>
> > (Q4) The formulation of the style coder is based on human heuristic, instead of data driven. How can you tell what styles can be controlled?
>
> Thank you for the good question. Since our encoders are trained fully unsupervised manner (i.e., no attribute label is used specifically for disentangling the encoders), they are not strictly independent. However, still they are trained to mainly encode different features of the image. It can be supported by three experiment results. First, we can see that the principal components of the style space are high-level attributes (Fig. 5 in the supplementary) while the principal components of the content space are more related to the spatial information, e.g., pose, and face size (Fig. 6 in the supplementary). Second, Table 4 in the main paper shows that the more style information is given ($\lambda=0.25$) the more attribute from the style image is applied (and thus less attribute from the content image is maintained). In other words, lining up with the PCA experiments, the classifier-based experiment shows that the style space mainly encodes high-level semantics. Third, by seeing Fig. 8 in the main paper and Fig. 19 in the supplementary, we can see that the structure of the building comes from the content image while the color and the texture of the building come from the style image.
>
> > (Q3) How does the proposed timestep-dependent weight schedule compare with existing methods in terms of computational efficiency and quality of generated images?
>
> The performance comparisons with/without timestep scheduling are reported in Table 3 and Fig 6. Without timestep scheduling, GCDM shows better performance in both FID (realism) and LPIPS (diversity) than CDM. Combined with timestep scheduling, both CDM and GCDM show meaningful improvements in FID in exchange for losing diversity. In other words, by introducing timestep scheduling, we can expect more realistic but less diverse results. Regarding computational efficiency, it is as efficient as the regular diffusion sampling process because the only added computation is to get a scalar weight $w_c$ or $w_s$ given a timestep $t$. Please see Section 3.2 in the main paper for more concrete descriptions.

---

### Official Review · Reviewer_HEZE · 2023-11-01

**Soundness:** 1 poor
**Presentation:** 2 fair
**Contribution:** 2 fair
**Rating:** 3
**Confidence:** 4

**Summary:**

The authors propose an improved strategy for generating conditional samples in the setting where the conditions and their composition is semantically complex.

**Strengths:**

- The proposed method appears to improve empirically on CDM on several displayed generation examples -- namely, when there is unclear composition of two disparate (and more rarely composed) conditions, such as an octopus next to a pyramid, and a bear with a car.

- The authors evaluate the proposed method on a selection of benchmarks, and appear to obtain improved performance over existing baselines.

**Weaknesses:**

- Poor formatting. Why are some equations not numbered in the main text? Some variables are never defined in the un-numbered equations. (See questions.)

- Unclear exposition. The concepts and notation in Sections 3.1 and 3.3 are difficult to reconcile. What is $\epsilon(x_t, t, c_1, c_2)$? Are $z_s$ and $z_c$ just the conditions $c_1$ and $c_2$, respectively? How is this a generalization of CDM?

- Limited novelty. It appears to me that the main contribution involves the addition of joint conditions, rather than simple independent conditions (CDM) in Eq. 3. This is overall a rather simple addition from CDM. Proposition 3.2 is not very interesting, as it appears to simply state that GCDM generalized CDM (which is known) and CDM generalized CFG (which is known).

- Lack of scalability. In the current formulation, it is not clear to me how the proposed method will scale well with increasing conditions. The number of joint conditions required in Eq. 3 seems to grow combinatorially with the number of conditions.

- Lack of ablation study. The authors propose several orthogonal improvements (e.g. timestep scheduling and an adaptive group normalization, a.k.a. AdaGN, layer). How much do these aspects contribute to the performance of the model?

- Significant increase in hyperparameters. It appears that at least 9 new hyperparameters are introduced in this conditioning method ($a$, $b$, $\alpha$, $\lambda$, $\beta_i$, $t_1$, $t_2$, $t_3$). It is not clear to me how to choose these hyperparameters, and to what extent the observed empirical improvements can be attributed to hand-tuning of these hyperparameters.

**Questions:**

What is $\epsilon_t$ in the un-numbered (first) equation in Section 3.1?

What is the intuition behind the basic form $(1 + t_1\phi(z_c))(1 + \zeta(z_s))((1 + t_2)h + t_3)$? Why is it basic?

How are the various new hyperparameters $a$, $b$, $\alpha$, $\lambda$, $\beta_i$, $t_1$, $t_2$, $t_3$ chosen? How robust are the results to perturbations in these hyperparameters?

---

> ### Author Response · Authors · 2023-11-22
> **Response to Reviewer HEZE**
>
> > (W1) Poor formatting. Why are some equations not numbered in the main text? Some variables are never defined in the un-numbered equations. (See questions.)
>
> > (Q1) What is $\epsilon_t$ in the un-numbered (first) equation in Section 3.1?
>
> We will make a revision by adding a number in the first equation and defining $\epsilon$ in the first equation. $\epsilon$ is a denoising UNet.
>
> > (W2) Unclear exposition. The concepts and notation in Sections 3.1 and 3.3 are difficult to reconcile. What is $\epsilon(x_t, t, c_1, c_2)$? Are $z_s$ and $z_c$ just the conditions $c_1$ and $c_2$, respectively? How is this a generalization of CDM?
>
> $\epsilon(x_t, t, c_1, c_2)$ is a denoising UNet taking two conditions. As mentioned right above Definition 3.1 in the main paper, $c_1$ and $c_2$ are $z_s$ and $z_c$, respectively. As for the question of how GCDM can generalize CDM, as shown in Eq. 5, given $\lambda=0$, GCDM formulation simplifies to CDM.
>
> > (W3) Limited novelty.
>
> As mentioned in the introduction of the main paper and overview of the supplementary material (Section A), our technical novelties are not limited to extending CDM to GCDM. We are proposing three methods for enhancing controllability of Diffusion Models; 1. a framework for having content and style latent space, 2. timestep scheduling during the sampling, 3. GCDM. Moreover, regarding a justification of GCDM, even though its form can be seen as a simple extension of CDM, we argue that the simplicity cannot be a lack of novelty if the extension makes something better that is meaningful and was not achievable beforehand. For example, GCDM gives more controllability to the Diffusion sampling algorithm by merging CDM and joint guidance in the form of a geometric average (which helps us to sample from both the joint and the non-joint terms while maintaining the amount of the conditional signal compared to the unconditional signal.) By doing so, we have shown that we can have one more dimension to control (i.e., realism) compared to CDM. The low FID score of the high $\lambda$ from Table 1 and Table 2 are quantitatively supporting our argument. Also, Fig. 5 is qualitatively verifying our argument.
>
> > (W4) Limited scalability
>
> Thank you for pointing out a valid point. As pointed out, naively extending GCDM to $N$ conditions requires computing a lot of terms, e.g., $\epsilon(x_t, t, c_1, c_2, ..., c_N), \epsilon(x_t, t, c_1, c_2, ..., c_{N-1}), \epsilon(x_t, t, c_1, c_2, ..., c_{N-2}), ...$. However, the scalability issue can be resolved by only computing the full joint and the single conditionals, i.e., $\epsilon(x_t, t, c_1, c_2, ..., c_N), \epsilon(x_t, t, c_1), \epsilon(x_t, t, c_2), ..., \epsilon(x_t, t, c_N)$. Thus, given $N$ conditions (where $N \geq 2$), Eq. (3) in the main paper turns to:
>
> $$
> \nabla_{x_t} \log \tilde{p}_{\alpha, \lambda, \beta_1, \beta_2, ..., \beta_N}(x_t|c_1,c_2, ... ,c_N) = \epsilon(x_t,t) + \alpha \left[ \lambda \Bigl( \epsilon(x_t, t, c_1,c_2, ..., c_N) - \epsilon(x_t, t) \Bigr) + (1-\lambda) \sum _{i=1}^N \beta_i \Bigl( \epsilon(x_t,t,c_i)- \epsilon(x_t,t) \Bigr) \right]
> $$
>
> Additionally, the computational cost of the changed form can be even reduced if we can have a prior, e.g., which condition would be more emphasized so that only a few single conditionals (out of $N$-single conditionals) and the full joint will be computed during the diffusion sampling process.
>
> > (W5) Lack of ablation study
>
> our technical novelties are 1. having two latent spaces, 2. timestep scheduling, 3. GCDM. The performance with/without timestep scheduling and the performance with/without GCDM are reported in Table 3 and Fig. 6 in the main paper. Details are briefly but clearly mentioned in "Effect of Timestep Scheduling" paragraph on page 8.
>
> Regarding the first novelty (having two latent spaces), its performance improvement can be seen by comparing DiffAE+SDEdit in Table 1 (single latent space) and CDM, w/o schedule in Table 3 (two latent spaces). As can be seen from the comparison, by setting two latent spaces, we can get better improvement in FID. Also, having two spaces gives us richer controllability as shown in Section F in the supplementary (please skim through the figures). Please note that we did not argue that the variant of AdaGN is our technical novelty.
>
> > (W6) Significant increase in hyperparameters
>
> There are five hyperparameters in our proposed sampling methods ($a, b$ from timestep scheduling and $\alpha, \lambda, \beta$ from GCDM). Please note that $t'$s are timestep embeddings. Since we agreed with the complexity of the hyperparameter tuning, we reported specific visual guidance on how to control each hyperparameter (in Fig. 1 in the supplementary). Please see the figure.
> Even though the number of hyperparameters can be complex in the first place, we believe it enables more subtle control for users to find the best output, as shown in the wide range of controllability in Fig. 1 in the supplementary.

---

### Meta-Review · Area_Chair_xRYR · 2023-12-08

**Metareview:**

This paper is on conditioning Diffusion models for image generation using latent codes for spatial content and style. This approach generates combinations of content/style images with better controllability. On strengths: the explicit disentangement of style and content leads to somewhat better interpretability, and the method achieves a good performance on FFHQ, AFHQ, LSUN datasets. On weaknesses: The implementation is rather complex with several additional hyperparameters; presentation issues and lack of coherent ablations were also raised in the reviews.

**Justification For Why Not Higher Score:**

Unclear exposition, limited novelty, not fully convincing experiments,  scalability concerns were all raised consistently as grounds for rejection.

**Justification For Why Not Lower Score:**

N/A

---

### Decision · Program_Chairs · 2024-01-16

Reject